# Calibration tests in multi-class classification:
# A unifying framework

**David Widmann**
Department of Information Technology
Uppsala University, Sweden
`david.widmann@it.uu.se`

**Fredrik Lindsten**
Division of Statistics and Machine Learning
Linköping University, Sweden
`fredrik.lindsten@liu.se`

**Dave Zachariah**
Department of Information Technology
Uppsala University, Sweden
`dave.zachariah@it.uu.se`

## Abstract

In safety-critical applications a probabilistic model is usually required to be calibrated, i.e., to capture the uncertainty of its predictions accurately. In multi-class classification, calibration of the most confident predictions only is often not sufficient. We propose and study calibration measures for multi-class classification that generalize existing measures such as the expected calibration error, the maximum calibration error, and the maximum mean calibration error. We propose and evaluate empirically different consistent and unbiased estimators for a specific class of measures based on matrix-valued kernels. Importantly, these estimators can be interpreted as test statistics associated with well-defined bounds and approximations of the p-value under the null hypothesis that the model is calibrated, significantly improving the interpretability of calibration measures, which otherwise lack any meaningful unit or scale.

## 1   Introduction

Consider the problem of analyzing microscopic images of tissue samples and reporting a tumour grade, i.e., a score that indicates whether cancer cells are well-differentiated or not, affecting both prognosis and treatment of patients. Since for some pathological images not even experienced pathologists might all agree on one classification, this task contains an inherent component of uncertainty. This type of uncertainty that can not be removed by increasing the size of the training data set is typically called aleatoric uncertainty (Kiureghian and Ditlevsen, 2009). Unfortunately, even if the ideal model is among the class of models we consider, with a finite training data set we will never obtain the ideal model but we can only hope to learn a model that is, in some sense, close to it. Worse still, our model might not even be close to the ideal model if the model class is too restrictive or the number of training data is small—which is not unlikely given the fact that annotating pathological images is expensive. Thus ideally our model should be able to express not only aleatoric uncertainty but also the uncertainty about the model itself. In contrast to aleatoric uncertainty this so-called epistemic uncertainty can be reduced by additional training data.

Dealing with these different types of uncertainty is one of the major problems in machine learning. The application of our model in clinical practice demands "meaningful" uncertainties to avoid doing harm to patients. Being too certain about high tumour grades might cause harm due to unneeded aggressive therapies and overly pessimistic prognoses, whereas being too certain about low tumour grades might result in insufficient therapies. "Proper" uncertainty estimates are also crucial if the

model is supervised by a pathologist that takes over if the uncertainty reported by the model is too high. False but highly certain gradings might incorrectly keep the pathologist out of the loop, and on the other hand too uncertain gradings might demand unneeded and costly human intervention.

Probability theory provides a solid framework for dealing with uncertainties. Instead of assigning exactly one grade to each pathological image, so-called probabilistic models report subjective probabilities, sometimes also called confidence scores, of the tumour grades for each image. The model can be evaluated by comparing these subjective probabilities to the ground truth.

One desired property of such a probabilistic model is sharpness (or high accuracy), i.e., if possible, the model should assign the highest probability to the true tumour grade (which maybe can not be inferred from the image at hand but only by other means such as an additional immunohistochemical staining). However, to be able to trust the predictions the probabilities should be calibrated (or reliable) as well (DeGroot and Fienberg, 1983; Murphy and Winkler, 1977). This property requires the subjective probabilities to match the relative empirical frequencies: intuitively, if we could observe a long run of predictions $(0.5, 0.1, 0.1, 0.3)$ for tumour grades $1, 2, 3$, and $4$, the empirical frequencies of the true tumour grades should be $(0.5, 0.1, 0.1, 0.3)$. Note that accuracy and calibration are two complementary properties: a model with over-confident predictions can be highly accurate but miscalibrated, whereas a model that always reports the overall proportion of patients of each tumour grade in the considered population is calibrated but highly inaccurate.

Research of calibration in statistics and machine learning literature has been focused mainly on binary classification problems or the most confident predictions: common calibration measures such as the expected calibration error (ECE) (Naeini et al., 2015), the maximum calibration error (MCE) (Naeini et al., 2015), and the kernel-based maximum mean calibration error (MMCE) (Kumar et al., 2018), and reliability diagrams (Murphy and Winkler, 1977) have been developed for binary classification. This is insufficient since many recent applications of machine learning involve multiple classes. Furthermore, the crucial finding of Guo et al. (2017) that many modern deep neural networks are miscalibrated is also based only on the most confident prediction.

Recently Vaicenavicius et al. (2019) suggested that this analysis might be too reduced for many realistic scenarios. In our example, a prediction of $(0.5, 0.3, 0.1, 0.1)$ is fundamentally different from a prediction of $(0.5, 0.1, 0.1, 0.3)$, since according to the model in the first case it is only half as likely that a tumour is of grade 3 or 4, and hence the subjective probability of missing out on a more aggressive therapy is smaller. However, commonly in the study of calibration all predictions with a highest reported confidence score of $0.5$ are grouped together and a calibrated model has only to be correct about the most confident tumour grade in 50% of the cases, regardless of the other predictions. Although the ECE can be generalized to multi-class classification, its applicability seems to be limited since its histogram-regression based estimator requires partitioning of the potentially high-dimensional probability simplex and is asymptotically inconsistent in many cases (Vaicenavicius et al., 2019). Sample complexity bounds for a bias-reduced estimator of the ECE introduced in metereological literature (Bröcker, 2011; Ferro and Fricker, 2012) were derived in concurrent work (Kumar et al., 2019).

## 2   Our contribution

In this work, we propose and study a general framework of calibration measures for multi-class classification. We show that this framework encompasses common calibration measures for binary classification such as the expected calibration error (ECE), the maximum calibration error (MCE), and the maximum mean calibration error (MMCE) by Kumar et al. (2018). In more detail we study a class of measures based on vector-valued reproducing kernel Hilbert spaces, for which we derive consistent and unbiased estimators. The statistical properties of the proposed estimators are not only theoretically appealing, but also of high practical value, since they allow us to address two main problems in calibration evaluation.

As discussed by Vaicenavicius et al. (2019), all calibration error estimates are inherently random, and comparing competing models based on these estimates without taking the randomness into account can be very misleading, in particular when the estimators are biased (which, for instance, is the case for the commonly used histogram-regression based estimator of the ECE). Even more fundamentally, all commonly used calibration measures lack a meaningful unit or scale and are therefore not interpretable as such (regardless of any finite sample issues).

The consistency and unbiasedness of the proposed estimators facilitate comparisons between competing models, and allow us to derive multiple statistical tests for calibration that exploit these properties. Moreover, by viewing the estimators as *calibration test statistics*, with well-defined bounds and approximations of the corresponding p-value, we give them an interpretable meaning.

We evaluate the proposed estimators and statistical tests empirically and compare them with existing methods. To facilitate multi-class calibration evaluation we provide the Julia packages `ConsistencyResampling.jl` (Widmann, 2019c), `CalibrationErrors.jl` (Widmann, 2019a), and `CalibrationTests.jl` (Widmann, 2019b) for consistency resampling, calibration error estimation, and calibration tests, respectively.

# 3 Background

We start by shortly summarizing the most relevant definitions and concepts. Due to space constraints and to improve the readability of our paper, we do not provide any proofs in the main text but only refer to the results in the supplementary material, which is intended as a reference for mathematically precise statements and proofs.

## 3.1 Probabilistic setting

Let $(X, Y)$ be a pair of random variables with $X$ and $Y$ representing inputs (features) and outputs, respectively. We focus on classification problems and hence without loss of generality we may assume that the outputs consist of the $m$ classes $1, \ldots, m$.

Let $\Delta^m$ denote the $(m - 1)$-dimensional probability simplex $\Delta^m := \{z \in \mathbb{R}^m_{\geq 0} : \|z\|_1 = 1\}$. Then a *probabilistic model* $g$ is a function that for every input $x$ outputs a prediction $g(x) \in \Delta^m$ that models the distribution

$$\big(\mathbb{P}[Y = 1 \,|\, X = x], \ldots, \mathbb{P}[Y = m \,|\, X = x]\big) \in \Delta^m$$

of class $Y$ given input $X = x$.

## 3.2 Calibration

### 3.2.1 Common notion

The common notion of calibration, as, e.g., used by Guo et al. (2017), considers only the most confident predictions $\max_y g_y(x)$ of a model $g$. According to this definition, a model is calibrated if

$$\mathbb{P}[Y = \arg\max_y g_y(X) \,|\, \max_y g_y(X)] = \max_y g_y(X) \tag{1}$$

holds almost always. Thus a model that is calibrated according to Eq. (1) ensures that we can *partly trust* the uncertainties reported by its predictions. As an example, for a prediction of $(0.4, 0.3, 0.3)$ the model would only guarantee that in the long run inputs that yield a most confident prediction of $40\%$ are in the corresponding class $40\%$ of the time.[1]

### 3.2.2 Strong notion

According to the more general calibration definition of Bröcker (2009); Vaicenavicius et al. (2019), a probabilistic model $g$ is calibrated if for almost all inputs $x$ the prediction $g(x)$ is equal to the distribution of class $Y$ given prediction $g(X) = g(x)$. More formally, a calibrated model satisfies

$$\mathbb{P}[Y = y \,|\, g(X)] = g_y(X) \tag{2}$$

almost always for all classes $y \in \{1, \ldots, m\}$. As Vaicenavicius et al. (2019) showed, for multi-class classification this formulation is stronger than the definition of Zadrozny and Elkan (2002) that only demands calibrated marginal probabilities. Thus we can *fully trust* the uncertainties reported by the predictions of a model that is calibrated according to Eq. (2). The prediction $(0.4, 0.3, 0.3)$ would actually imply that the class distribution of the inputs that yield this prediction is $(0.4, 0.3, 0.3)$. To emphasize the difference to the definition in Eq. (1), we call calibration according to Eq. (2) *calibration in the strong sense* or *strong calibration*.

To simplify our notation, we rewrite Eq. (2) in vectorized form. Equivalent to the definition above, a model $g$ is calibrated in the strong sense if

$$r(g(X)) - g(X) = 0 \qquad (3)$$

holds almost always, where

$$r(\xi) := \big( \mathbb{P}[Y = 1 \mid g(X) = \xi], \ldots, \mathbb{P}[Y = m \mid g(X) = \xi] \big)$$

is the distribution of class $Y$ given prediction $g(X) = \xi$.

The calibration of certain aspects of a model, such as the five largest predictions or groups of classes, can be investigated by studying the strong calibration of models induced by so-called calibration lenses. For more details about evaluation and visualization of strong calibration we refer to Vaicenavicius et al. (2019).

### 3.3 Matrix-valued kernels

The miscalibration measure that we propose in this work is based on matrix-valued kernels $k \colon \Delta^m \times \Delta^m \to \mathbb{R}^{m \times m}$. Matrix-valued kernels can be defined in a similar way as the more common real-valued kernels, which can be characterized as symmetric positive definite functions (Berlinet and Thomas-Agnan, 2004, Lemma 4).

**Definition 1 (Micchelli and Pontil (2005, Definition 2); Caponnetto et al. (2008, Definition 1)).** We call a function $k \colon \Delta^m \times \Delta^m \to \mathbb{R}^{m \times m}$ a *matrix-valued kernel* if for all $s, t \in \Delta^m$ $k(s,t) = k(t,s)^\mathsf{T}$ and it is positive semi-definite, i.e., if for all $n \in \mathbb{N}$, $t_1, \ldots, t_n \in \Delta^m$, and $u_1, \ldots, u_n \in \mathbb{R}^m$

$$\sum_{i,j=1}^{n} u_i^\mathsf{T} k(t_i, t_j) u_j \geq 0.$$

There exists a one-to-one mapping of such kernels and reproducing kernel Hilbert spaces (RKHSs) of vector-valued functions $f \colon \Delta^m \to \mathbb{R}^m$. We provide a short summary of RKHSs of vector-valued functions on the probability simplex in Appendix D. A more detailed general introduction to RKHSs of vector-valued functions can be found in the publications by Caponnetto et al. (2008); Carmeli et al. (2010); Micchelli and Pontil (2005).

Similar to the scalar case, matrix-valued kernels can be constructed from other matrix-valued kernels and even from real-valued kernels. Very simple matrix-valued kernels are kernels of the form $k(s,t) = \tilde{k}(s,t)\mathbf{I}_m$, where $\tilde{k}$ is a scalar-valued kernel, such as the Gaussian or Laplacian kernel, and $\mathbf{I}_m$ is the identity matrix. As Example D.1 shows, this construction can be generalized by, e.g., replacing the identity matrix with an arbitrary positive semi-definite matrix.

An important class of kernels are so-called universal kernels. Loosely speaking, a kernel is called universal if its RKHS is a dense subset of the space of continuous functions, i.e., if in the neighbourhood of every continuous function we can find a function in the RKHS. Prominent real-valued kernels on the probability simplex such as the Gaussian and the Laplacian kernel are universal, and can be used to construct universal matrix-valued kernels of the form in Example D.1, as Lemma D.3 shows.

## 4 Unification of calibration measures

In this section we introduce a general measure of strong calibration and show its relation to other existing measures.

### 4.1 Calibration error

In the analysis of strong calibration the discrepancy in the left-hand side of Eq. (3) lends itself naturally to the following calibration measure.

**Definition 2.** Let $\mathcal{F}$ be a non-empty space of functions $f \colon \Delta^m \to \mathbb{R}^m$. Then the calibration error (CE) of model $g$ with respect to class $\mathcal{F}$ is

$$\mathrm{CE}[\mathcal{F}, g] := \sup_{f \in \mathcal{F}} \mathbb{E}\left[ (r(g(X)) - g(X))^\mathsf{T} f(g(X)) \right].$$

A trivial consequence of the design of the CE is that the measure is zero for every model that is calibrated in the strong sense, regardless of the choice of $\mathcal{F}$. The converse statement is not true in general. As we show in Theorem C.2, the class of continuous functions is a space for which the CE is zero if and only if model $g$ is strongly calibrated, and hence allows to characterize calibrated models. However, since this space is extremely large, for every model the CE is either 0 or $\infty$.[2] Thus a measure based on this space does not allow us to compare miscalibrated models and hence is rather impractical.

## 4.2 Kernel calibration error

Due to the correspondence between kernels and RKHSs we can define the following kernel measure.

**Definition 3.** Let $k$ be a matrix-valued kernel as in Definition 1. Then we define the kernel calibration error (KCE) with respect to kernel $k$ as $\mathrm{KCE}[k, g] \coloneqq \mathrm{CE}[\mathcal{F}, g]$, where $\mathcal{F}$ is the unit ball in the RKHS corresponding to kernel $k$.

As mentioned above, a RKHS with a universal kernel is a dense subset of the space of continuous functions. Hence these kernels yield a function space that is still large enough for identifying strongly calibrated models.

**Theorem 1 (cf. Theorem C.1).** *Let $k$ be a matrix-valued kernel as in Definition 1, and assume that $k$ is universal. Then $\mathrm{KCE}[k, g] = 0$ if and only if model $g$ is calibrated in the strong sense.*

From the supremum-based Definition 2 it might not be obvious how the KCE can be evaluated. Fortunately, there exists an equivalent kernel-based formulation.

**Lemma 1 (cf. Lemma E.2).** *Let $k$ be a matrix-valued kernel as in Definition 1. If $\mathbb{E}[\|k(g(X), g(X))\|] < \infty$, then*

$$\mathrm{KCE}[k, g] = \left( \mathbb{E} \left[ (e_Y - g(X))^\mathsf{T} k(g(X), g(X'))(e_{Y'} - g(X')) \right] \right)^{1/2}, \tag{4}$$

*where $(X', Y')$ is an independent copy of $(X, Y)$ and $e_i$ denotes the $i$th unit vector.*

## 4.3 Expected calibration error

The most common measure of calibration error is the expected calibration error (ECE). Typically it is used for quantifying calibration in a binary classification setting but it generalizes to strong calibration in a straightforward way. Let $d \colon \Delta^m \times \Delta^m \to \mathbb{R}_{\geq 0}$ be a distance measure on the probability simplex. Then the expected calibration error of a model $g$ with respect to $d$ is defined as

$$\mathrm{ECE}[d, g] = \mathbb{E}[d(r(g(X)), g(X))]. \tag{5}$$

If $d(p, q) = 0 \Leftrightarrow p = q$, as it is the case for standard choices of $d$ such as the total variation distance or the (squared) Euclidean distance, then $\mathrm{ECE}[d, g]$ is zero if and only if $g$ is strongly calibrated as per Eq. (3).

The ECE with respect to the cityblock distance, the total variation distance, or the squared Euclidean distance, are special cases of the calibration error CE, as we show in Lemma I.1.

## 4.4 Maximum mean calibration error

Kumar et al. (2018) proposed a kernel-based calibration measure, the so-called maximum mean calibration error (MMCE), for training (better) calibrated neural networks. In contrast to their work, in our publication we do not discuss how to obtain calibrated models but focus on the evaluation of calibration and on calibration tests. Moreover, the MMCE applies only to a binary classification setting whereas our measure quantifies strong calibration and hence is more generally applicable. In fact, as we show in Example I.1, the MMCE is a special case of the KCE.

# 5 Calibration error estimators

Consider the task of estimating the calibration error of model $g$ using a validation set $\mathcal{D} = \{(X_i, Y_i)\}_{i=1}^n$ of $n$ i.i.d. random pairs of inputs and labels that are distributed according to $(X, Y)$.

From the expression for the ECE in Eq. (5), the natural (and, indeed, standard) approach for estimating the ECE is as the sample average of the distance $d$ between the predictions $g(X)$ and the calibration function $r(g(X))$. However, this is problematic since the calibration function is not readily available and needs to be estimated as well. Typically, this is addressed using histogram-regression, see, e.g., Guo et al. (2017); Naeini et al. (2015); Vaicenavicius et al. (2019), which unfortunately leads to inconsistent and biased estimators in many cases (Vaicenavicius et al., 2019) and can scale poorly to large $m$. In contrast, for the KCE in Eq. (4) there is no explicit dependence on $r$, which enables us to derive multiple consistent and also unbiased estimators.

Let $k$ be a matrix-valued kernel as in Definition 1 with $\mathbb{E}[\|k(g(X), g(X))\|] < \infty$, and define for $1 \leq i, j \leq n$
$$h_{i,j} := (e_{Y_i} - g(X_i))^{\mathsf{T}} k(g(X_i), g(X_j))(e_{Y_j} - g(X_j)).$$

Then the estimators listed in Table 1 are consistent estimators of the squared kernel calibration error $\mathrm{SKCE}[k, g] := \mathrm{KCE}^2[k, g]$ (see Lemmas F.1 to F.3). The subscript letters "q" and "l" refer to the quadratic and linear computational complexity of the unbiased estimators, respectively.

Table 1: Three consistent estimators of the SKCE.

| Notation | Definition | Properties | Complexity |
|---|---|---|---|
| $\widehat{\mathrm{SKCE}}_{\mathrm{b}}$ | $n^{-2} \sum_{i,j=1}^n h_{i,j}$ | biased | $O(n^2)$ |
| $\widehat{\mathrm{SKCE}}_{\mathrm{uq}}$ | $\binom{n}{2}^{-1} \sum_{1 \leq i < j \leq n} h_{i,j}$ | unbiased | $O(n^2)$ |
| $\widehat{\mathrm{SKCE}}_{\mathrm{ul}}$ | $\lfloor n/2 \rfloor^{-1} \sum_{i=1}^{\lfloor n/2 \rfloor} h_{2i-1,2i}$ | unbiased | $O(n)$ |

# 6 Calibration tests

In general, calibration errors do not have a meaningful unit or scale. This renders it difficult to interpret an estimated non-zero error and to compare different models. However, by viewing the estimates as test statistics with respect to Eq. (3), they obtain an interpretable probabilistic meaning.

Similar to the derivation of the two-sample tests by Gretton et al. (2012), we can use the consistency and unbiasedness of the estimators of the SKCE presented above to obtain bounds and approximations of the p-value for the null hypothesis $H_0$ that the model is calibrated, i.e., for the probability that the estimator is greater than or equal to the observed calibration error estimate, if the model is calibrated. These bounds and approximations do not only allow us to perform hypothesis testing of the null hypothesis $H_0$, but they also enable us to transfer unintuitive calibration error estimates to an intuitive and interpretable probabilistic setting.

As we show in Theorems H.2 to H.4, we can obtain so-called distribution-free bounds without making any assumptions about the distribution of $(X, Y)$ or the model $g$. A downside of these uniform bounds is that usually they provide only a loose bound of the p-value.

**Lemma 2 (Distribution-free bounds (see Theorems H.2 to H.4)).** *Let $k$ be a matrix-valued kernel as in Definition 1, and assume that $K_{p;q} := \sup_{s,t \in \Delta^m} \|k(s,t)\|_{p;q} < \infty$ for some $1 \leq p, q \leq \infty$.[3] Let $t > 0$ and $B_{p;q} := 2^{1+1/p-1/q} K_{p;q}$, then for the biased estimator we can bound*

$$\mathbb{P}\left[\widehat{\mathrm{SKCE}}_{\mathrm{b}} \geq t \,\middle|\, H_0\right] \leq \exp\left(-\frac{1}{2}\left(\max\left\{0, \sqrt{nt/B_{p;q}} - 1\right\}\right)^2\right),$$

*and for either of the unbiased estimators $T \in \{\widehat{\mathrm{SKCE}}_{\mathrm{uq}}, \widehat{\mathrm{SKCE}}_{\mathrm{ul}}\}$, we can bound*

$$\mathbb{P}\left[T \geq t \,\middle|\, H_0\right] \leq \exp\left(-\frac{\lfloor n/2 \rfloor t^2}{2 B_{p;q}^2}\right).$$

Asymptotic bounds exploit the asymptotic distribution of the test statistics under the null hypothesis, as the number of validation data points goes to infinity. The central limit theorem implies that the linear estimator is asymptotically normally distributed.

**Lemma 3 (Asymptotic distribution of $\widehat{\mathrm{SKCE}}_{\mathrm{ul}}$ (see Corollary G.1)).** *Let $k$ be a matrix-valued kernel as in Definition 1, and assume that $\mathbb{E}[\|k(g(X), g(X))\|] < \infty$. If $\mathrm{Var}[h_{i,j}] < \infty$, then*

$$\mathbb{P}\left[\sqrt{\lfloor n/2 \rfloor}\, \widehat{\mathrm{SKCE}}_{\mathrm{ul}} \geq t \,\Big|\, H_0\right] \to 1 - \Phi\left(\frac{t}{\hat{\sigma}}\right) \quad as \quad n \to \infty,$$

*where $\hat{\sigma}$ is the sample standard deviation of $h_{2i-1,2i}$ $(i = 1, \ldots, \lfloor n/2 \rfloor)$ and $\Phi$ is the cumulative distribution function of the standard normal distribution.*

In Theorem G.2 we derive a theoretical expression of the asymptotic distribution of $n\,\widehat{\mathrm{SKCE}}_{\mathrm{uq}}$, under the assumption of strong calibration. This limit distribution can be approximated, e.g., by bootstrapping (Arcones and Giné, 1992) or Pearson curve fitting, as discussed by Gretton et al. (2012).

## 7 Experiments

We conduct experiments to confirm the derived theoretical properties of the proposed calibration error estimators empirically and to compare them with a standard histogram-regression based estimator of the ECE, denoted by $\widehat{\mathrm{ECE}}$.[4]

We construct synthetic data sets $\{(g(X_i), Y_i)\}_{i=1}^{250}$ of 250 labeled predictions for $m = 10$ classes from three generative models. For each model we first sample predictions $g(X_i) \sim \mathrm{Dir}(0.1, \ldots, 0.1)$, and then simulate corresponding labels $Y_i$ conditionally on $g(X_i)$ from

$$\mathbf{M1}\colon\ \mathrm{Cat}(g(X_i)), \quad \mathbf{M2}\colon\ 0.5\,\mathrm{Cat}(g(X_i)) + 0.5\,\mathrm{Cat}(1, 0, \ldots, 0), \quad \mathbf{M3}\colon\ \mathrm{Cat}(0.1, \ldots, 0.1),$$

where **M1** gives a calibrated model, and **M2** and **M3** are uncalibrated. In Appendix J.2 we investigate the theoretical properties of these models in more detail.

For simplicity, we use the matrix-valued kernel $k(x, y) = \exp\left(-\|x - y\|/\nu\right)\mathbf{I}_{10}$, where the kernel bandwidth $\nu > 0$ is chosen by the median heuristic. The total variation distance is a common distance measure of probability distributions and the standard distance measure for the ECE (Bröcker and Smith, 2007; Guo et al., 2017; Vaicenavicius et al., 2019), and hence it is chosen as the distance measure for all studied calibration errors.

### 7.1 Calibration error estimates

In Fig. 1 we show the distribution of $\widehat{\mathrm{ECE}}$ and of the three proposed estimators of the SKCE, evaluated on $10^4$ randomly sampled data sets from each of the three models. The true calibration error of these models, indicated by a dashed line, is calculated theoretically for the ECE (see Appendix J.2.1) and empirically for the SKCE using the sample mean of all unbiased estimates of $\widehat{\mathrm{SKCE}}_{\mathrm{uq}}$.

We see that the standard estimator of the ECE exhibits both negative and positive bias, whereas $\widehat{\mathrm{SKCE}}_{\mathrm{b}}$ is theoretically guaranteed to be biased upwards. The results also confirm the unbiasedness of $\widehat{\mathrm{SKCE}}_{\mathrm{ul}}$.

### 7.2 Calibration tests

We repeatedly compute the bounds and approximations of the p-value for the calibration error estimators that were derived in Section 6 on $10^4$ randomly sampled data sets from each of the three models. More concretely, we evaluate the distribution-free bounds for $\widehat{\mathrm{SKCE}}_{\mathrm{b}}$ ($\mathbf{D}_{\mathrm{b}}$), $\widehat{\mathrm{SKCE}}_{\mathrm{uq}}$ ($\mathbf{D}_{\mathrm{uq}}$), and $\widehat{\mathrm{SKCE}}_{\mathrm{ul}}$ ($\mathbf{D}_{\mathrm{ul}}$) and the asymptotic approximations for $\widehat{\mathrm{SKCE}}_{\mathrm{uq}}$ ($\mathbf{A}_{\mathrm{uq}}$) and $\widehat{\mathrm{SKCE}}_{\mathrm{ul}}$ ($\mathbf{A}_{\mathrm{l}}$), where the former is approximated by bootstrapping. We compare them with a previously proposed hypothesis test for the standard ECE estimator based on consistency resampling ($\mathbf{C}$), in which

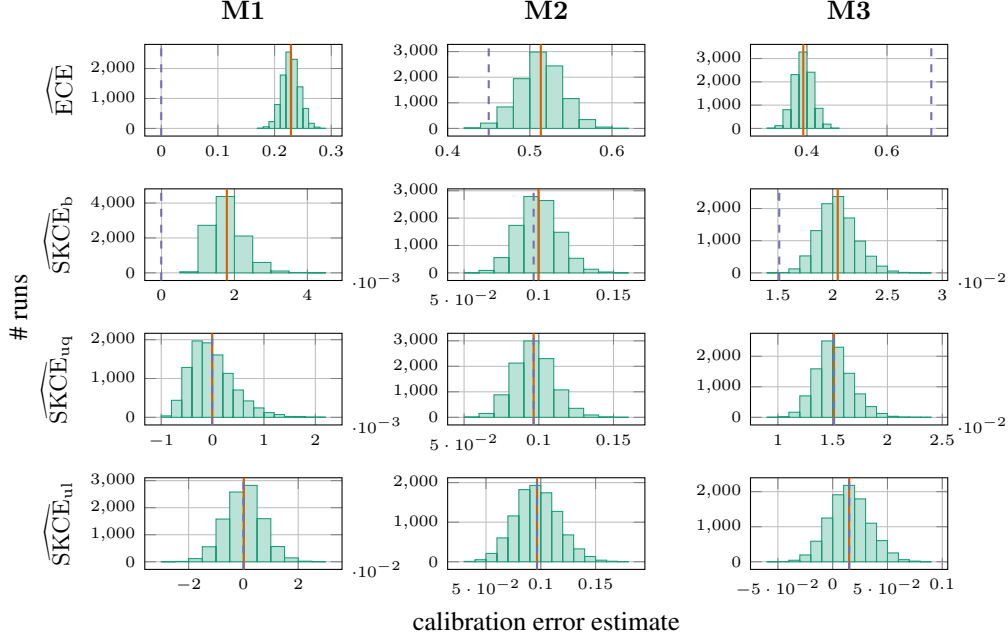

Figure 1: Distribution of calibration error estimates of $10^4$ data sets that are randomly sampled from the generative models **M1**, **M2**, and **M3**. The solid line indicates the mean of the calibration error estimates, and the dashed line displays the true calibration error.

data sets are resampled under the assumption that the model is calibrated by sampling labels from resampled predictions (Bröcker and Smith, 2007; Vaicenavicius et al., 2019).

For a chosen significance level $\alpha$ we compute from the p-value approximations $p_1, \ldots, p_{10^4}$ the empirical test error

$$\frac{1}{10^4} \sum_{i=1}^{10^4} \mathbb{1}_{[0,\alpha]}(p_i) \quad \text{(for } \mathbf{M1)} \quad \text{and} \quad \frac{1}{10^4} \sum_{i=1}^{10^4} \mathbb{1}_{(\alpha,1]}(p_i) \quad \text{(for } \mathbf{M2} \text{ and } \mathbf{M3).}$$

In Fig. 2 we plot these empirical test errors versus the significance level.

As expected, the distribution-free bounds seem to be very loose upper bounds of the p-value, resulting in statistical tests without much power. The asymptotic approximations, however, seem to estimate the p-value quite well on average, as can be seen from the overlap with the diagonal in the results for the calibrated model **M1** (the empirical test error matches the chosen significance level). Additionally, calibration tests based on asymptotic distribution of these statistics, and in particular of $\widehat{\mathrm{SKCE}}_{\mathrm{uq}}$, are quite powerful in our experiments, as the results for the uncalibrated models **M2** and **M3** show. For the calibrated model, consistency resampling leads to an empirical test error that is not upper bounded by the significance level, i.e., the null hypothesis of the model being calibrated is rejected too often. This behaviour is caused by an underestimation of the p-value on average, which unfortunately makes the calibration test based on consistency resampling for the standard ECE estimator unreliable.

### 7.3 Additional experiments

In Appendix J.2.3 we provide additional results for varying number of classes and a non-standard ECE estimator with data-dependent bins. We observe that the bias of $\widehat{\mathrm{ECE}}$ becomes more prominent with increasing number of classes, showing high calibration error estimates even for calibrated models. The estimators of the SKCE are not affected by the number of classes in the same way. In some experiments with 100 and 1000 classes, however, the distribution of $\widehat{\mathrm{SKCE}}_{\mathrm{ul}}$ shows multi-modality.

The considered calibration measures depend only on the predictions and the true labels, not on how these predictions are computed. Hence directly specifying the predictions allows a clean numerical

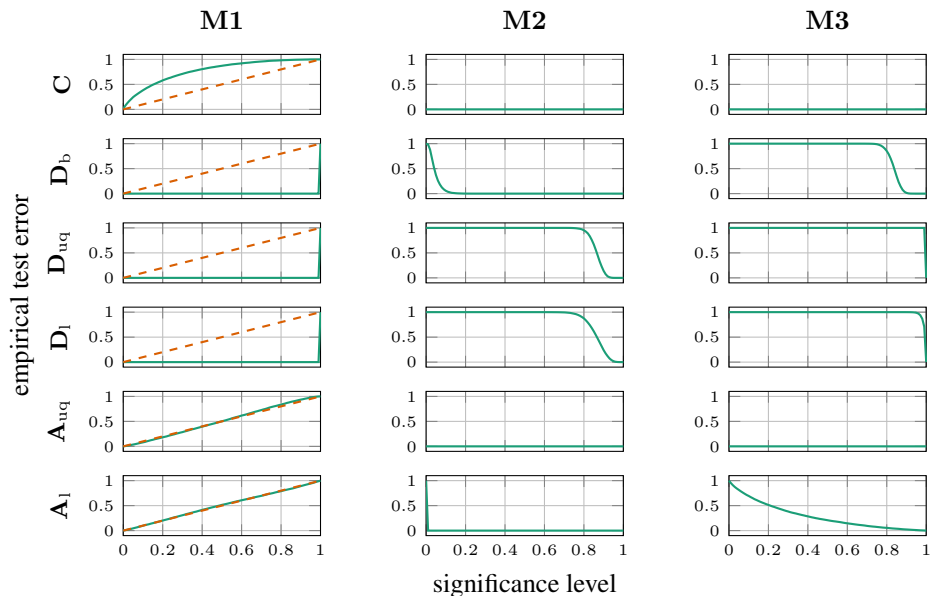

Figure 2: Empirical test error versus significance level for different bounds and approximations of the p-value, evaluated on $10^4$ data sets that are randomly sampled from the generative models **M1**, **M2**, and **M3**. The dashed line highlights the diagonal of the unit square.

evaluation and enables comparisons of the estimates with the true calibration error. Nevertheless, we provide a more practical evaluation of calibration for several modern neural networks in Appendix J.3.

# 8 Conclusion

We have presented a unified framework for quantifying the calibration error of probabilistic classifiers. The framework encompasses several existing error measures and enables the formulation of a new kernel-based measure. We have derived unbiased and consistent estimators of the kernel-based error measures, which are properties not readily enjoyed by the more common and less tractable ECE. The impact of the kernel and its hyperparameters on the estimators is an important question for future research. We have refrained from investigating it in this paper, since it deserves a more exhaustive study than, what we felt, would have been possible in this work.

The calibration error estimators can be viewed as test statistics. This confers probabilistic interpretability to the error measures. Specifically, we can compute well-founded bounds and approximations of the p-value for the observed error estimates under the null hypothesis that the model is calibrated. We have derived distribution-free bounds and asymptotic approximations for the estimators of the proposed kernel-based error measure, that allow reliable calibration tests in contrast to previously proposed tests based on consistency resampling with the standard estimator of the ECE.

**Acknowledgements**

We thank the reviewers for all the constructive feedback on our paper. This research is financially supported by the Swedish Research Council via the projects *Learning of Large-Scale Probabilistic Dynamical Models* (contract number: 2016-04278) and *Counterfactual Prediction Methods for Heterogeneous Populations* (contract number: 2018-05040), by the Swedish Foundation for Strategic Research via the project *Probabilistic Modeling and Inference for Machine Learning* (contract number: ICA16-0015), and by the Wallenberg AI, Autonomous Systems and Software Program (WASP) funded by the Knut and Alice Wallenberg Foundation.

## Footnotes

[1]This notion of calibration does not consider for which class the most confident prediction was obtained.

[2]Assume $\mathrm{CE}[\mathcal{F}, g] < \infty$ and let $f_1, f_2, \ldots$ be a sequence of continuous functions with $\mathrm{CE}[\mathcal{F}, g] = \lim_{n \to \infty} \mathbb{E}\left[(r(g(X)) - g(X))^\mathsf{T} f_n(g(X))\right]$. From Remark C.2 we know that $\mathrm{CE}[\mathcal{F}, g] \geq 0$. Moreover, $\tilde{f}_n \coloneqq 2f_n$ are continuous functions with $2\,\mathrm{CE}[\mathcal{F}, g] = \lim_{n \to \infty} \mathbb{E}\left[(r(g(X)) - g(X))^\mathsf{T} \tilde{f}_n(g(X))\right] \leq \sup_{f \in \mathcal{F}} \mathbb{E}\left[(r(g(X)) - g(X))^\mathsf{T} f(g(X))\right] = \mathrm{CE}[\mathcal{F}, g]$. Thus $\mathrm{CE}[\mathcal{F}, g] = 0$.

[3]For a matrix $A$ we denote by $\|A\|_{p;q}$ the induced matrix norm $\sup_{x \neq 0} \|Ax\|_q / \|x\|_p$.

[4]The implementation of the experiments is available online at `https://github.com/devmotion/CalibrationPaper`.

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
