[Supplementary Material]

## A   Notation

We introduce some additional notation to keep the following discussion concise.

Let $U$ be a compact metric space and $V$ be a Hilbert space. In this paper we only consider $U = [0, 1]$ and $U = \Delta^m$. By $\mathcal{C}(U, V)$ we denote the space of continuous functions $f\colon U \to V$.

For $1 \le p < \infty$, $L^p(U, \mu; V)$ is the space of (equivalence classes of) measurable functions $f\colon U \to V$ such that $\|f\|^p$ is $\mu$-integrable, equipped with norm $\|f\|_{\mu,p} = \left( \int_U \|f(x)\|^p \, \mu(\mathrm{d}x) \right)^{1/p}$; for $p = \infty$, $L^\infty(U, \mu; V)$ is the space of $\mu$-essentially bounded measurable functions $f\colon U \to V$ with norm $\|f\|_{\mu,\infty} = \mu\text{-}\operatorname{ess\,sup}_{x \in \mathcal{X}} \|f(x)\|$. We denote the closed unit ball of the space $L^p(U, \mu; V)$ by $K^p(U, \mu; V) \coloneqq \{ f \in L^p(U, \mu; V) \colon \|f\|_{\mu,p} \le 1 \}$.

If the norm $\|.\|_V$ on $V$ is not clear from the context we indicate it by writing $L^p(U, \mu; V, \|.\|_V)$, $K^p(U, \mu; V, \|.\|_V)$, and $\|.\|_{\mu,p;\|.\|_V}$. If $V \subset \mathbb{R}^d$, all possible norms $\|.\|_V$ are equivalent and hence we choose $\|.\|_p$ on $V$ to simplify our calculations, if not stated otherwise. Moreover, if $V \subset \mathbb{R}^d$ and $\|.\|_V = \|.\|_q$ for some $1 \le q \le \infty$, for convenience we write $\|.\|_{\mu,p;q} = \|.\|_{\mu,p;\|.\|_V}$.

Let $W$ be another Hilbert space. Then $\mathcal{L}(V, W)$ denotes the space of bounded linear operators $T\colon V \to W$; if $W = V$, we write $\mathcal{L}(V) \coloneqq \mathcal{L}(V, V)$. The induced operator norm on $\mathcal{L}(V, W)$ is defined by

$$
\begin{aligned}
\|T\|_{\|.\|_V; \|.\|_W} &= \inf \{ c \ge 0 \colon \|Tv\|_W \le c \|v\|_V \text{ for all } v \in V \} \\
&= \sup_{v \in V \colon \|v\|_V \le 1} \|Tv\|_W.
\end{aligned}
$$

If $W = V$, we write $\|.\|_{\|.\|_V} = \|.\|_{\|.\|_V; \|.\|_V}$. Moreover, for convenience if $V \subset \mathbb{R}^d$ and $\|.\|_V = \|.\|_p$ for some $1 \le p \le \infty$, we use the notation $\|.\|_{p; \|.\|_W} = \|.\|_{\|.\|_V; \|.\|_W}$. In the same way, if $W \subset \mathbb{R}^d$ and $\|.\|_W = \|.\|_q$ for some $1 \le q \le \infty$, we write $\|.\|_{\|.\|_V; q} = \|.\|_{\|.\|_V; \|.\|_W}$.

By $\mathcal{B}(T)$ we denote the Borel $\sigma$-algebra of a topological space $T$.

We write $\mu_A$ for the distribution of a random variable $A$, i.e., the pushforward measure it induces, if $A$ is defined on a probability space with probability measure $\mu$.

## B   Probabilistic setting

Let $(\Omega, \mathcal{A}, \mu)$ be a probability space. Let $m \in \mathbb{N}$ and define the random variables $X\colon (\Omega, \mathcal{A}) \to (\mathcal{X}, \Sigma_X)$ and $Y\colon (\Omega, \mathcal{A}) \to (\{1, \ldots, m\}, 2^{\{1, \ldots, m\}})$, such that $\Sigma_X$ contains all singletons. We denote a version of the regular conditional distribution of $Y$ given $X$ by $\mu_{Y|X}(\cdot|x)$ for all $x \in \mathcal{X}$.

We consider the problem of learning a measurable function $g\colon (\mathcal{X}, \Sigma_X) \to (\Delta^m, \mathcal{B}(\Delta^m))$ that returns the prediction $g_y(x)$ of $\mu_{Y|X}(\{y\}|x)$ for all $x \in \mathcal{X}$ and $y \in \{1, \ldots, m\}$. We define the random variable $G\colon (\Omega, \mathcal{A}) \to (\Delta^m, \mathcal{B}(\Delta^m))$ as $G \coloneqq g(X)$.

In the same way as above, we denote a version of the regular conditional distribution of $Y$ given $G$ by $\mu_{Y|G}(\cdot|t)$ for all $t \in \Delta^m$. The function $\delta\colon (\Delta^m, \mathcal{B}(\Delta^m)) \to (\mathbb{R}^m, \mathcal{B}(\mathbb{R}^m))$, given by

$$
\delta(t) \coloneqq \begin{pmatrix} \mu_{Y|G}(1|t) - t \\ \vdots \\ \mu_{Y|G}(m|t) - t \end{pmatrix}
$$

for all $t \in \Delta^m$, gives rise to another random variable $\Delta\colon (\Omega, \mathcal{A}) \to (\mathbb{R}^m, \mathcal{B}(\mathbb{R}^m))$ that is defined by $\Delta \coloneqq \delta(G)$.

Using the newly introduced mathematical notation, we can reformulate strong calibration in a compact way. A model $g$ is calibrated in the strong sense if $\mu_{Y|G}(\cdot|G) = G$ almost surely, or equivalently if $\Delta = 0$ almost surely.

## C  Calibration error

**Definition C.1 (Calibration error).** Let $\mathcal{F} \subset L^1(\Delta^m, \mu_G; \mathbb{R}^m)$ be non-empty. Then the calibration error CE of model $g$ with respect to class $\mathcal{F}$ is

$$\mathrm{CE}[\mathcal{F}, g] := \sup_{f \in \mathcal{F}} \mathbb{E}\left[\langle \Delta, f(G) \rangle_{\mathbb{R}^m}\right].$$

*Remark* C.1. Note that $\|\Delta\|_\infty \leq 1$ almost surely, and thus $\|\delta\|_{\mu_G, \infty} \leq 1$. Hence by Hölder's inequality for all $f \in \mathcal{F}$ we have

$$|\mathbb{E}[\langle \Delta, f(G) \rangle_{\mathbb{R}^m}]| \leq \mathbb{E}[|\langle \Delta, f(G) \rangle_{\mathbb{R}^m}|] \leq \|\delta\|_{\mu_G, \infty} \|f\|_{\mu_G, 1} \leq \|f\|_{\mu_G, 1} < \infty.$$

However, it is still possible that $\mathrm{CE}[\mathcal{F}, g] = \infty$.

*Remark* C.2. If $\mathcal{F}$ is symmetric in the sense that $f \in \mathcal{F}$ implies $-f \in \mathcal{F}$, then $\mathrm{CE}[\mathcal{F}, g] \geq 0$.

The measure highly depends on the choice of $\mathcal{F}$ but strong calibration always implies a calibration error of zero.

**Theorem C.1 (Strong calibration implies zero error).** *Let $\mathcal{F} \subset L^1(\Delta^m, \mu_G; \mathbb{R})$. If model $g$ is calibrated in the strong sense, then $\mathrm{CE}[\mathcal{F}, g] = 0$.*

*Proof.* If model $g$ is calibrated in the strong sense, then $\Delta = 0$ almost surely. Hence for all $f \in \mathcal{F}$ we have $\mathbb{E}[\langle \Delta, f(G) \rangle_{\mathbb{R}}] = 0$, which implies $\mathrm{CE}[\mathcal{F}, g] = 0$. $\qquad\square$

Of course, the converse statement is not true in general. A similar result as above shows that the class of continuous functions, albeit too large and impractical, allows to identify calibrated models.

**Theorem C.2.** *Let $\mathcal{F} = \mathcal{C}(\Delta^m, \mathbb{R}^m)$. Then $\mathrm{CE}[\mathcal{F}, g] = 0$ if and only if model $g$ is calibrated in the strong sense.*

*Proof.* Note that $\mathcal{F}$ is well defined since $\mathcal{F} \subset L^1(\Delta^m, \mu_G; \mathbb{R}^m)$.

If model $g$ is calibrated in the strong sense, then $\mathrm{CE}[\mathcal{F}, g] = 0$ by Theorem C.1.

If model $g$ is not calibrated in the strong sense, then $\Delta = 0$ does not hold almost surely. In particular, there exists $s \in \{-1, 1\}^m$ such that $\langle \Delta, s \rangle_{\mathbb{R}^m} \leq 0$ does not hold almost surely. Define the function $f_s \colon \Delta^m \to \mathbb{R}^m$ by $f_s := \langle \delta(\cdot), s \rangle_{\mathbb{R}^m}$ and let $A_s := f_s^{-1}((0, \infty))$. Then $A_s \in \mathcal{B}(\Delta^m)$ since $f_s$ is Borel measurable, and $\mu_G(A_s) > 0$. Hence we know that

$$\alpha_s := \mathbb{E}[\langle \Delta, s \mathbb{1}_{A_s}(G) \rangle_{\mathbb{R}^m}] > 0.$$

Since $\mu_G$ is a Borel probability measure on a compact metric space, it is regular and hence there exist a compact set $K$ and an open set $U$ such that $K \subset A_s \subset U$ and $\mu_G(U \setminus K) < \alpha_s/4$ (Rudin, 1986, Theorem 2.17). Thus by Urysohn's lemma applied to the closed sets $K$ and $U^c$, there exists a continuous function $h \in \mathcal{C}(\Delta^m, \mathbb{R})$ such that $\mathbb{1}_K \leq h \leq \mathbb{1}_U$. By defining $f = sh \in \mathcal{C}(\Delta^m, \mathbb{R}^m)$ and applying Hölder's inequality we obtain

$$\begin{aligned}
\mathbb{E}[\langle \Delta, f(G) \rangle_{\mathbb{R}^m}] &= \mathbb{E}[\langle \Delta, s \mathbb{1}_{A_s}(G) \rangle_{\mathbb{R}^m}] \\
&\quad + \mathbb{E}[\langle \Delta, f(G) - s \mathbb{1}_{A_s}(G) \rangle_{\mathbb{R}^m}] \\
&\geq \alpha_s - |\mathbb{E}[(h(G) - \mathbb{1}_{A_s}(G)) \langle \Delta, s \rangle_{\mathbb{R}^m}]| \\
&\geq \alpha_s - \mathbb{E}[|h(G) - \mathbb{1}_{A_s}(G)| |\langle \Delta, s \rangle_{\mathbb{R}^m}|] \\
&\geq \alpha_s - \mathbb{E}[\mathbb{1}_{U \setminus K}(G) \|\Delta\|_1 \|s\|_\infty] \geq \alpha_s - 2\mu_G(U \setminus K) \\
&> \alpha_s - \alpha_s/2 = \alpha_s/2 > 0.
\end{aligned}$$

This implies $\mathrm{CE}[\mathcal{F}, g] > 0$. $\qquad\square$

## D  Reproducing kernel Hilbert spaces of vector-valued functions on the probability simplex

**Definition D.1 (Micchelli and Pontil (2005, Definition 1)).** Let $\mathcal{H}$ be a Hilbert space of vector-valued functions $f \colon \Delta^m \to \mathbb{R}^m$ with inner product $\langle ., . \rangle_{\mathcal{H}}$. We call $\mathcal{H}$ a reproducing kernel Hilbert space (RKHS), if for all $t \in \Delta^m$ and $u \in \mathbb{R}^m$ the functional $E_{t,u} \colon \mathcal{H} \to \mathbb{R}$, $E_{t,u} f := \langle u, f(t) \rangle_{\mathbb{R}^m}$, is a bounded (or equivalently continuous) linear operator.

Riesz's representation theorem ensures that there exists a unique function $k \colon \Delta^m \times \Delta^m \to \mathbb{R}^{m \times m}$ such that for all $t \in \Delta^m$ the function $k(\cdot, t)$ is a linear map from $\Delta^m$ to $\mathcal{H}$ and for all $u \in \mathbb{R}^m$ it satisfies the so-called reproducing property

$$\langle u, f(t) \rangle_{\mathbb{R}^m} = E_{t,u} f = \langle k(\cdot, t)u, f \rangle_{\mathcal{H}}. \tag{D.1}$$

It can be shown that function $k$ is self-adjoint[5] and positive semi-definite, and hence a kernel according to Definition 1 (Micchelli and Pontil, 2005, Proposition 1). Similar to the scalar-valued case, conversely by the Moore-Aronszajn theorem (Aronszajn, 1950) to every kernel $k \colon \Delta^m \times \Delta^m \to \mathcal{L}(\mathbb{R}^m)$ there exists a unique RKHS $\mathcal{H} \subset (\mathbb{R}^m)^{\Delta^m}$ with $k$ as reproducing kernel (Micchelli and Pontil, 2005, Theorem 1).

Other useful properties are summarized in Lemma D.1. Micchelli and Pontil (2005) considered only the Euclidean norm on $\mathbb{R}^m$, corresponding to $p = q = 2$ in our statement. For convenience we use the notation $\|.\|_{p;\mathcal{H}} = \|.\|_{\|.\|_p; \|.\|_{\mathcal{H}}}$.

**Lemma D.1 (Micchelli and Pontil (2005, Proposition 1)).** *Let $\mathcal{H} \subset (\mathbb{R}^m)^{\Delta^m}$ be a RKHS with kernel $k \colon \Delta^m \times \Delta^m \to \mathcal{L}(\mathbb{R}^m)$. Let $1 \le p, q \le \infty$ with Hölder conjugates $p'$ and $q'$, respectively.*

1. *For all $t \in \Delta^m$*

$$\|k(\cdot, t)\|_{p;\mathcal{H}} = \|k(t,t)\|_{p;p'}^{1/2}. \tag{D.2}$$

2. *For all $s, t \in \Delta^m$*

$$\|k(s,t)\|_{p;q} \le \|k(s,s)\|_{q';q}^{1/2} \|k(t,t)\|_{p;p'}^{1/2}. \tag{D.3}$$

3. *For all $f \in \mathcal{H}$ and $t \in \Delta^m$*

$$\|f(t)\|_p \le \|f\|_{\mathcal{H}} \|k(\cdot, t)\|_{p';\mathcal{H}} = \|f\|_{\mathcal{H}} \|k(t,t)\|_{p';p}^{1/2}. \tag{D.4}$$

*Proof.* Let $t \in \Delta^m$. From the reproducing property, Hölder's inequality, and the definition of the operator norm, we obtain for all $u \in \mathbb{R}^m$

$$\|k(\cdot,t)u\|_{\mathcal{H}}^2 = \langle k(\cdot,t)u, k(\cdot,t)u \rangle_{\mathcal{H}} = \langle u, k(t,t)u \rangle_{\mathbb{R}^m} \le \|u\|_p \|k(t,t)u\|_{p'} \le \|u\|_p^2 \|k(t,t)\|_{p;p'},$$

which implies that

$$\|k(\cdot,t)\|_{p;\mathcal{H}} = \sup_{u \in \mathbb{R}^m \setminus \{0\}} \frac{\|k(\cdot,t)u\|_{\mathcal{H}}}{\|u\|_p} \le \|k(t,t)\|_{p;p'}^{1/2}. \tag{D.5}$$

On the other hand, for all $u, v \in \mathbb{R}^m$ it follows from the reproducing property, the Cauchy-Schwarz inequality, and the definition of the operator norm that

$$\langle u, k(t,t)v \rangle_{\mathbb{R}^m} = \langle k(\cdot,t)u, k(\cdot,t)v \rangle_{\mathcal{H}} \le \|k(\cdot,t)u\|_{\mathcal{H}} \|k(\cdot,t)v\|_{\mathcal{H}} \le \|u\|_p \|v\|_p \|k(\cdot,t)\|_{p;\mathcal{H}}^2.$$

Since the $\ell_p$-norm is the dual norm of the $\ell_{p'}$-norm, it follows that

$$\|k(t,t)v\|_{p'} = \sup_{u \in \mathbb{R}^m \colon \|u\|_p \le 1} \langle u, k(t,t)v \rangle_{\mathbb{R}^m} \le \|v\|_p \|k(\cdot,t)\|_{p;\mathcal{H}}^2,$$

which implies that

$$\|k(t,t)\|_{p;p'} = \sup_{v \in \mathbb{R}^m \setminus \{0\}} \frac{\|k(t,t)v\|_{p'}}{\|v\|_p} \le \|k(\cdot,t)\|_{p;\mathcal{H}}^2. \tag{D.6}$$

Equation (D.2) follows from Eqs. (D.5) and (D.6).

Let $s, t \in \Delta^m$. From the reproducing property, the Cauchy-Schwarz inequality, and the definition of the operator norm, we get for all $u, v \in \mathbb{R}^m$

$$\langle u, k(s,t)v \rangle_{\mathbb{R}^m} \le \langle k(\cdot,s)u, k(\cdot,t)v \rangle_{\mathcal{H}} \le \|k(\cdot,s)u\|_{\mathcal{H}} \|k(\cdot,t)v\|_{\mathcal{H}}$$
$$\le \|u\|_{q'} \|v\|_p \|k(\cdot,s)\|_{q';\mathcal{H}} \|k(\cdot,t)\|_{p;\mathcal{H}}.$$

Thus we obtain

$$\|k(s,t)v\|_q = \sup_{u\in\mathbb{R}^m:\,\|u\|_{q'}\leq 1}\langle u, k(s,t)v\rangle_{\mathbb{R}^m} \leq \|v\|_p\|k(\cdot,s)\|_{q';\mathcal{H}}\|k(\cdot,t)\|_{p;\mathcal{H}},$$

which implies

$$\|k(s,t)\|_{p;q} = \sup_{v\in\mathbb{R}^m\setminus\{0\}}\frac{\|k(s,t)v\|_q}{\|v\|_p} \leq \|k(\cdot,s)\|_{q';\mathcal{H}}\|k(\cdot,t)\|_{p;\mathcal{H}}.$$

Hence from Eq. (D.2) we obtain Eq. (D.3).

For the third statement, let $f\in\mathcal{H}$ and $t\in\Delta^m$. From the reproducing property, the Cauchy-Schwarz inequality, and the definition of the operator norm, we obtain for all $u\in\mathbb{R}^m$

$$\langle u, f(t)\rangle_{\mathbb{R}^m} = \langle k(\cdot,t)u, f\rangle_{\mathcal{H}} \leq \|k(\cdot,t)u\|_{\mathcal{H}}\|f\|_{\mathcal{H}} \leq \|u\|_{p'}\|k(\cdot,t)\|_{p';\mathcal{H}}\|f\|_{\mathcal{H}}.$$

Thus the duality of the $\ell_p$- and the $\ell_{p'}$-norm implies

$$\|f(t)\|_p = \sup_{u\in\mathbb{R}^m:\,\|u\|_{p'}\leq 1}\langle u, f(t)\rangle_{\mathbb{R}^m} \leq \|k(\cdot,t)\|_{p';\mathcal{H}}\|f\|_{\mathcal{H}},$$

which together with Eq. (D.2) yields Eq. (D.4). $\qquad\square$

If $\mu$ is a measure on $\Delta^m$, we define for $1\leq p,q\leq\infty$ $\|k\|_{\mu,p;q} := \|\tilde{k}_q\|_{\mu,p}$ where $\tilde{k}_q\colon\Delta^m\to\mathbb{R}$ is given by $\tilde{k}(t) := \|k(\cdot,t)\|_{q;\mathcal{H}} = \|k(t,t)\|_{q;q'}^{1/2}$. We omit the value of $q$ if it is clear from the context or does not matter, since all norms on $\mathbb{R}^m$ are equivalent.

It is possible to construct certain classes of matrix-valued kernels from scalar-valued kernels, as the following example shows.

**Example D.1 (Micchelli and Pontil (2005); Caponnetto et al. (2008, Example 1))**
For all $i\in\{1,\ldots,n\}$, let $k_i\colon\Delta^m\times\Delta^m\to\mathbb{R}$ be a scalar-valued kernel and $A_i\in\mathbb{R}^{m\times m}$ be a positive semi-definite matrix. Then the function

$$k\colon\Delta^m\times\Delta^m\to\mathbb{R}^{m\times m}, \qquad k(s,t) := \sum_{i=1}^n k_i(s,t)A_i, \tag{D.7}$$

is a matrix-valued kernel.

We state a simple result about measurability of functions in the considered RKHSs. The result can be formulated in a much more general fashion and is similar to a result by Christmann and Steinwart (2008, Lemma 4.24).

**Lemma D.2 (Measurable RKHS).** *Let $\mathcal{H}\subset\left(\mathbb{R}^m\right)^{\Delta^m}$ be a RKHS with kernel $k\colon\Delta^m\times\Delta^m\to\mathcal{L}(\mathbb{R}^m)$. Then all $f\in\mathcal{H}$ are measurable if and only if $k(\cdot,t)u\in\left(\mathbb{R}^m\right)^{\Delta^m}$ is measurable for all $t\in\Delta^m$ and $u\in\mathbb{R}^m$.*

*Proof.* If all $f\in\mathcal{H}$ are measurable, then $k(\cdot,t)u\in\mathcal{H}$ is measurable for all $t\in\Delta^m$ and $u\in\mathbb{R}^m$.

If $k(\cdot,t)u$ is measurable for all $t\in\Delta^m$ and $u\in\mathbb{R}^m$, then all functions in $\mathcal{H}_0 := \operatorname{span}\{k(\cdot,t)u\colon t\in\Delta^m, u\in\mathbb{R}^m\}\subset\mathcal{H}$ are measurable.

Let $f\in\mathcal{H}$. Since $\mathcal{H}=\overline{\mathcal{H}}_0$ (see, e.g., Carmeli et al., 2010), there exists a sequence $(f_n)_{n\in\mathbb{N}}\subset\mathcal{H}_0$ such that $\lim_{n\to\infty}\|f-f_n\|_{\mathcal{H}}=0$. For all $t\in\Delta^m$, since the operator $k^*(\cdot,t)$ is continuous, by the reproducing property we obtain $\lim_{n\to\infty}f_n(t)=f(t)$. Thus $f$ is measurable. $\qquad\square$

By definition (see, e.g., Carmeli et al., 2010, Definition 1), a RKHS with a continuous kernel is a subspace of the space of continuous functions. The following equivalent formulation is an immediate consequence of the result by Carmeli et al. (2010).

**Corollary D.1 (Carmeli et al. (2010, Proposition 1)).** *A kernel $k\colon\Delta^m\times\Delta^m\to\mathbb{R}^m$ is continuous if for all $t\in\Delta^m$ $t\mapsto\|k(t,t)\|$ is bounded and for all $t\in\Delta^m$ and $u\in\mathbb{R}^m$ $k(\cdot,t)u$ is a continuous function from $\Delta^m$ to $\mathbb{R}^m$.*

An important class of continuous kernels are so-called universal kernels, for which the corresponding RKHS is a dense subset of the space of continuous functions with respect to the uniform norm. A result by Caponnetto et al. (2008) shows under what assumptions matrix-valued kernels of the form in Example D.1 are universal.

**Lemma D.3** (Caponnetto et al. (2008, Theorem 14)). *For all $i \in \{1, \ldots, n\}$, let $k_i \colon \Delta^m \times \Delta^m \to \mathbb{R}$ be a universal scalar-valued kernel and $A_i \in \mathbb{R}^{m \times m}$ be a positive semi-definite matrix. Then the matrix-valued kernel defined in Eq. (D.7) is universal if and only if $\sum_{i=1}^n A_i$ is positive definite.*

# E  Kernel calibration error

The one-to-one correspondence between matrix-valued kernels and RKHSs of vector-valued functions motivates the introduction of the kernel calibration error (KCE) in Definition 3. For certain kernels we are able to identify strongly calibrated models.

**Theorem E.1.** *Let $k \colon \Delta^m \times \Delta^m \to \mathcal{L}(\mathbb{R}^m)$ be a universal continuous kernel. Then $\mathrm{KCE}[k, g] = 0$ if and only if model $g$ is calibrated in the strong sense.*

*Proof.* Let $\mathcal{F}$ be the unit ball in the RKHS $\mathcal{H} \subset (\mathbb{R}^m)^{\Delta^m}$ corresponding to kernel $k$. Since kernel $k$ is continuous, by definition $\mathcal{H} \subset \mathcal{C}(\Delta^m, \mathbb{R}^m)$ (Carmeli et al., 2010, Definition 1). Thus $\mathcal{F}$ is well defined since $\mathcal{F} \subset \mathcal{C}(\Delta^m, \mathbb{R}^m) \subset L^1(\Delta^m, \mu_G; \mathbb{R}^m)$.

If $g$ is calibrated in the strong sense, it follows from Theorem C.1 that $\mathrm{KCE}[k, g] = \mathrm{CE}[\mathcal{F}, g] = 0$.

Assume that $\mathrm{KCE}[k, g] = \mathrm{CE}[\mathcal{F}, g] = 0$. This implies $\mathbb{E}[\langle \Delta, f(G) \rangle_{\mathbb{R}^m}] = 0$ for all $f \in \mathcal{F}$. Let $f \in \mathcal{C}(\Delta^m, \mathbb{R}^m)$. Since $\mathcal{H}$ is dense in $\mathcal{C}(\Delta^m, \mathbb{R}^m)$ (Carmeli et al., 2010, Theorem 1), for all $\epsilon > 0$ there exists a function $h \in \mathcal{H}$ with $\|f - h\|_\infty < \epsilon/2$. Define $\tilde{h} \in \mathcal{F}$ by $\tilde{h} := h/\|h\|_{\mathcal{H}}$ if $\|h\|_{\mathcal{H}} \neq 0$ and $\tilde{h} := h$ otherwise. Since

$$\mathbb{E}[\langle \Delta, h(G) \rangle_{\mathbb{R}^m}] = \|h\|_{\mathcal{H}} \, \mathbb{E}[\langle \Delta, \tilde{h}(G) \rangle_{\mathbb{R}^m}] = 0,$$

by Hölder's inequality we obtain

$$\begin{aligned}
|\mathbb{E}[\langle \Delta, f(G) \rangle_{\mathbb{R}^m}]| &= |\mathbb{E}[\langle \Delta, f(G) - h(G) \rangle_{\mathbb{R}^m}]| \\
&\leq \mathbb{E}[|\langle \Delta, f(G) - h(G) \rangle_{\mathbb{R}^m}|] \\
&\leq \|\delta\|_{\mu_G, 1} \|f - h\|_{\mu_G, \infty} \\
&\leq 2\|f - h\|_\infty < \epsilon.
\end{aligned}$$

Thus $\mathrm{CE}[\mathcal{C}(\Delta^m, \mathbb{R}^m), g] = 0$, and hence $g$ is calibrated in the strong sense by Theorem C.2. □

Similar to the maximum mean discrepancy (Gretton et al., 2012), if we consider functions in a RKHS, we can rewrite the expectation $\mathbb{E}[\langle \Delta, f(G) \rangle_{\mathbb{R}^m}]$ as an inner product in the Hilbert space.

**Lemma E.1** (Existence and uniqueness of embedding). *Let $\mathcal{H} \subset (\mathbb{R}^m)^{\Delta^m}$ be a RKHS with kernel $k \colon \Delta^m \times \Delta^m \to \mathcal{L}(\mathbb{R}^m)$, and assume that $k(\cdot, t)u$ is measurable for all $t \in \Delta^m$ and $u \in \mathbb{R}^m$, and $\|k\|_{\mu_G, 1} < \infty$.*

*Then there exists a unique embedding $\mu_g \in \mathcal{H}$ such that for all $f \in \mathcal{H}$*

$$\mathbb{E}[\langle \Delta, f(G) \rangle_{\mathbb{R}^m}] = \langle f, \mu_g \rangle_{\mathcal{H}}.$$

*The embedding $\mu_g$ satisfies for all $t \in \Delta^m$ and $y \in \mathbb{R}^m$*

$$\langle y, \mu_g(t) \rangle_{\mathbb{R}^m} = \mathbb{E}[\langle \Delta, k(G, t)y \rangle_{\mathbb{R}^m}].$$

*Proof.* By Lemma D.2 all $f \in \mathcal{H}$ are measurable. Moreover, by Eq. (D.4) for all $f \in \mathcal{H}$ we have

$$\int_{\Delta^m} \|f(t)\|_1 \mu_G(\mathrm{d}t) \leq \|f\|_{\mathcal{H}} \int_{\Delta^m} \|k(t, t)\|_{\infty; 1}^{1/2} \mu_G(\mathrm{d}t)$$
$$= \|f\|_{\mathcal{H}} \|k\|_{\mu_G, 1; \infty} < \infty,$$

and thus $\mathcal{H} \subset L^1(\Delta^m, \mu_G; \mathbb{R}^m)$. Hence from Remark C.1 (with $\mathcal{F} = \mathcal{H}$) we know that for all $f \in \mathcal{H}$ the expectation $\mathbb{E}[\langle \Delta, f(G) \rangle_{\mathbb{R}^m}]$ exists and is finite.

Define the linear operator $T_g \colon \mathcal{H} \to \mathbb{R}$ by $T_g f := \mathbb{E}[\langle \Delta, f(G) \rangle_{\mathbb{R}^m}]$ for all $f \in \mathcal{H}$. In the same way as above, for all functions $f \in \mathcal{H}$ Hölder's inequality and Eq. (D.4) imply

$$
\begin{aligned}
|T_g f| = |\mathbb{E}[\langle \Delta, f(G) \rangle_{\mathbb{R}^m}] & \leq \mathbb{E}[|\langle \Delta, f(G) \rangle_{\mathbb{R}^m}|] \\
& \leq \|\delta\|_{\mu_G, \infty} \|f\|_{\mu_G, 1} \\
& \leq \|f\|_{\mu_G, 1} \leq \|f\|_{\mathcal{H}} \|k\|_{\mu_G, 1; \infty} < \infty.
\end{aligned}
$$

Thus $T_g$ is a continuous linear operator, and therefore it follows from Riesz's representation theorem that there exists a unique function $\mu_g \in \mathcal{H}$ such that

$$
\mathbb{E}[\langle \Delta, f(G) \rangle_{\mathbb{R}^m}] = T_g f = \langle f, \mu_g \rangle_{\mathcal{H}}
$$

for all $f \in \mathcal{H}$. This implies that for all $t \in \Delta^m$ and $y \in \mathbb{R}^m$

$$
\langle y, \mu_g(t) \rangle_{\mathbb{R}^m} = \langle k(\cdot, t) y, \mu_g \rangle_{\mathcal{H}} = \mathbb{E}[\langle \Delta, k(G, t) y \rangle_{\mathbb{R}^m}]. \qquad \square
$$

Lemma E.1 allows us to rewrite $\mathrm{KCE}[k, g]$ in a more explicit way.

**Lemma E.2 (Explicit formulation).** *Let $\mathcal{H} \subset (\mathbb{R}^m)^{\Delta^m}$ be a RKHS with kernel $k \colon \Delta^m \times \Delta^m \to \mathcal{L}(\mathbb{R}^m)$, and assume that $k(\cdot, t) u$ is measurable for all $t \in \Delta^m$ and $u \in \mathbb{R}^m$ and $\|k\|_{\mu_G, 1} < \infty$. Then*

$$
\mathrm{KCE}[k, g] = \|\mu_g\|_{\mathcal{H}},
$$

*where $\mu_g$ is the embedding defined in Lemma E.1. Moreover,*

$$
\mathrm{SKCE}[k, g] := \mathrm{KCE}^2[k, g] = \mathbb{E}[\langle e_Y - g(X), k(g(X), g(X'))(e_{Y'} - g(X')) \rangle_{\mathbb{R}^m}],
$$

*where $(X', Y')$ is an independent copy of $(X, Y)$ and $e_i$ denotes the $i$th unit vector.*

*Proof.* Let $\mathcal{F}$ be the unit ball in the RKHS $\mathcal{H}$. From Lemma E.1 we know that for all $f \in \mathcal{F}$ the expectation $\mathbb{E}[\langle \Delta, f(G) \rangle_{\mathbb{R}^m}]$ exists and is given by

$$
\mathbb{E}[\langle \Delta, f(G) \rangle_{\mathbb{R}^m}] = \langle f, \mu_g \rangle_{\mathcal{H}},
$$

where $\mu_g$ is the embedding defined in Lemma E.1. Thus the definition of the dual norm yields

$$
\mathrm{KCE}[k, g] = \mathrm{CE}[\mathcal{F}, g] = \sup_{f \in \mathcal{F}} \mathbb{E}[\langle \Delta, f(G) \rangle_{\mathbb{R}^m}] = \sup_{f \in \mathcal{F}} \langle f, \mu_g \rangle_{\mathcal{H}} = \|\mu_g\|_{\mathcal{H}}.
$$

Thus from the reproducing property and Lemma E.1 we obtain

$$
\begin{aligned}
\mathrm{SKCE}[k, g] = \mathrm{KCE}^2[k, g] = \langle \mu_g, \mu_g \rangle_{\mathcal{H}} & = \mathbb{E}[\langle \Delta, \mu_g(G) \rangle_{\mathbb{R}^m}] \\
& = \mathbb{E}[\mathbb{E}[\langle \Delta', k(G', G) \Delta \rangle_{\mathbb{R}^m} | G]] \\
& = \mathbb{E}[\langle \Delta, k(G, G') \Delta' \rangle_{\mathbb{R}^m}],
\end{aligned}
$$

where $G'$ is an independent copy of $G$ and $\Delta' := \delta(G')$.

By rewriting

$$
\Delta = \mathbb{E}[e_Y | G] - G = \mathbb{E}[e_Y - G | G]
$$

and $\Delta'$ in the same way, we get

$$
\mathrm{SKCE}[k, g] = \mathbb{E}[\langle e_Y - G, k(G, G')(e_{Y'} - G') \rangle_{\mathbb{R}^m}].
$$

Plugging in the definitions of $G$ and $G'$ yields

$$
\mathrm{SKCE}[k, g] = \mathbb{E}[\langle e_Y - g(X), k(g(X), g(X'))(e_{Y'} - g(X')) \rangle_{\mathbb{R}^m}]. \qquad \square
$$

## F  Estimators

Let $\mathcal{D} = \{(X_i, Y_i)\}_{i=1}^n$ be a set of random variables that are i.i.d. as $(X, Y)$. Regardless of the space $\mathcal{F}$ the plug-in estimator of $\mathrm{CE}[\mathcal{F}, g]$ is

$$
\widehat{\mathrm{CE}}[\mathcal{F}, g, \mathcal{D}] := \sup_{f \in \mathcal{F}} \frac{1}{n} \sum_{i=1}^n \langle \delta(X_i, Y_i), f(g(X_i)) \rangle_{\mathbb{R}^m}.
$$

If $\mathcal{F}$ is the unit ball in a RKHS, i.e., for the kernel calibration error, we can calculate this estimator explicitly.

**Lemma F.1 (Biased estimator).** *Let $\mathcal{F}$ be the unit ball in a RKHS $\mathcal{H} \subset (\mathbb{R}^m)^{\Delta^m}$ with kernel $k \colon \Delta^m \times \Delta^m \to \mathcal{L}(\mathbb{R}^m)$. Then*

$$\widehat{\mathrm{CE}}[\mathcal{F}, g, \mathcal{D}] = \frac{1}{n} \left[ \sum_{i,j=1}^{n} \langle \delta(X_i, Y_i), k(g(X_i), g(X_j)) \delta(X_j, Y_j) \rangle_{\mathbb{R}^m} \right]^{1/2}.$$

*Proof.* From the reproducing property and the definition of the dual norm it follows that

$$\widehat{\mathrm{CE}}[\mathcal{F}, g, \mathcal{D}] = \sup_{f \in \mathcal{F}} \left\langle \frac{1}{n} \sum_{i=1}^{n} k(\cdot, g(X_i)) \delta(X_i, Y_i), f \right\rangle_{\mathcal{H}} = \frac{1}{n} \left\| \sum_{i=1}^{n} k(\cdot, g(X_i)) \delta(X_i, Y_i) \right\|_{\mathcal{H}}.$$

Applying the reproducing property yields the result. $\qquad\square$

Since we can uniquely identify the unit ball $\mathcal{F}$ with the matrix-valued kernel $k$ and the plug-in estimator in Lemma F.1 does not depend on $\mathcal{F}$ explicitly, we introduce the notation

$$\widehat{\mathrm{KCE}}[k, g, \mathcal{D}] \coloneqq \widehat{\mathrm{CE}}[\mathcal{F}, g, \mathcal{D}] \quad \text{and} \quad \widehat{\mathrm{SKCE}}_{\mathrm{b}}[k, g, \mathcal{D}] \coloneqq \widehat{\mathrm{KCE}}^2[k, g, \mathcal{D}],$$

where $\mathcal{F}$ is the unit ball in the RKHS $\mathcal{H} \subset (\mathbb{R}^m)^{\Delta^m}$ corresponding to kernel $k$. By removing the terms involving the same random variables we obtain an unbiased estimator.

**Lemma F.2 (Unbiased estimator).** *Let $k \colon \Delta^m \times \Delta^m \to \mathcal{L}(\mathbb{R}^m)$ be a kernel, and assume that $k(\cdot, t)u$ is measurable for all $t \in \Delta^m$ and $u \in \mathbb{R}^m$, and $\|k\|_{\mu_G, 1} < \infty$. Then*

$$\widehat{\mathrm{SKCE}}_{\mathrm{uq}}[k, g, \mathcal{D}] \coloneqq \frac{1}{n(n-1)} \sum_{\substack{i,j=1, \\ i \neq j}}^{n} \langle \delta(X_i, Y_i), k(g(X_i), g(X_j)) \delta(X_j, Y_j) \rangle_{\mathbb{R}^m}$$

*is an unbiased estimator of* $\mathrm{SKCE}[k, g]$.

*Proof.* The assumptions of Lemma E.2 are satisfied, and hence we know that

$$\mathrm{SKCE}[k, g] = \mathbb{E}[\langle \delta(X, Y), k(g(X), g(X')) \delta(X', Y') \rangle_{\mathbb{R}^m}],$$

where $(X', Y')$ is an independent copy of $(X, Y)$. Since $(X, Y)$, $(X', Y')$, and $(X_i, Y_i)$ are i.i.d., we have

$$
\begin{aligned}
\mathbb{E}[\widehat{\mathrm{SKCE}}_{\mathrm{uq}}[k, g, \mathcal{D}]] &= \frac{1}{n(n-1)} \sum_{\substack{i=1, \\ i \neq j}}^{n} \mathbb{E}[\langle \delta(X, Y), k(g(X), g(X')) \delta(X', Y') \rangle_{\mathbb{R}^m}] \\
&= \mathbb{E}[\langle \delta(X, Y), k(g(X), g(X')) \delta(X', Y') \rangle_{\mathbb{R}^m}] \\
&= \mathrm{SKCE}[k, g],
\end{aligned}
$$

which shows that $\widehat{\mathrm{SKCE}}_{\mathrm{uq}}[k, g, \mathcal{D}]$ is an unbiased estimator of $\mathrm{SKCE}[k, g]$. $\qquad\square$

There exists an unbiased estimator with higher variance that scales not quadratically but only linearly with the number of samples.

**Lemma F.3 (Linear estimator).** *Let $k \colon \Delta^m \times \Delta^m \to \mathcal{L}(\mathbb{R}^m)$ be a kernel, and assume that $k(\cdot, t)u$ is measurable for all $t \in \Delta^m$ and $u \in \mathbb{R}^m$, and $\|k\|_{\mu_G, 1} < \infty$. Then*

$$\widehat{\mathrm{SKCE}}_{\mathrm{ul}}[k, g, \mathcal{D}] \coloneqq \frac{1}{\lfloor n/2 \rfloor} \sum_{i=1}^{\lfloor n/2 \rfloor} \langle \delta(X_{2i-1}, Y_{2i-1}), k(g(X_{2i-1}), g(X_{2i})) \delta(X_{2i}, Y_{2i}) \rangle_{\mathbb{R}^m}$$

*is an unbiased estimator of* $\mathrm{SKCE}[k, g]$.

*Proof.* The assumptions of Lemma E.2 are satisfied, and hence we know that

$$\mathrm{SKCE}[k, g] = \mathbb{E}[\langle \delta(X, Y), k(g(X), g(X'))\delta(X', Y')\rangle_{\mathbb{R}^m}],$$

where $(X', Y')$ is an independent copy of $(X, Y)$. Since $(X, Y)$, $(X', Y')$, and $(X_i, Y_i)$ are i.i.d., we have

$$
\begin{aligned}
\mathbb{E}[\widehat{\mathrm{SKCE}}_{\mathrm{ul}}[k, g, \mathcal{D}]] &= \frac{1}{\lfloor n/2 \rfloor} \sum_{i=1}^{\lfloor n/2 \rfloor} \mathbb{E}[\langle \delta(X, Y), k(g(X), g(X'))\delta(X', Y')\rangle_{\mathbb{R}^m}] \\
&= \mathbb{E}[\langle \delta(X, Y), k(g(X), g(X'))\delta(X', Y')\rangle_{\mathbb{R}^m}] \\
&= \mathrm{SKCE}[k, g],
\end{aligned}
$$

which shows that $\widehat{\mathrm{SKCE}}_{\mathrm{ul}}[k, g, \mathcal{D}]$ is an unbiased estimator of $\mathrm{SKCE}[k, g]$. $\qquad\square$

## G  Asymptotic distributions

In this section we investigate the asymptotic behaviour of the proposed estimators. We start with a simple but very useful statement.

**Lemma G.1.** *Let $k \colon \Delta^m \times \Delta^m \to \mathcal{L}(\mathbb{R}^m)$ be a kernel, and assume that $k(\cdot, t)u$ is measurable for all $t \in \Delta^m$ and $u \in \mathbb{R}^m$, and $\|k\|_{\mu_G, 2} < \infty$.*

*Then $\mathrm{Var}[\langle \Delta, k(G, G')\Delta'\rangle_{\mathbb{R}^m}] < \infty$, where $G'$ is an independent copy of $G$ and $\Delta' \coloneqq \delta(G')$.*

*Proof.* From the Cauchy-Schwarz inequality and the definition of the operator norm we obtain

$$\mathbb{E}[\langle \Delta, k(G, G')\Delta'\rangle_{\mathbb{R}^m}^2] \leq \mathbb{E}[\|\Delta\|_2^2 \|k(G, G')\|_{2;2}^2 \|\Delta\|_2^2] \leq 4\,\mathbb{E}[\|k(G, G')\|_{2;2}^2].$$

Hence by Eq. (D.3)

$$
\begin{aligned}
\mathbb{E}[\langle \Delta', k(G, G')\Delta'\rangle_{\mathbb{R}^m}^2] &\leq 4\,\mathbb{E}[\|k(G, G)\|_{2;2}\|k(G', G')\|_{2;2}] \\
&= 4\,\mathbb{E}[\|k(G, G)\|_{2;2}]\,\mathbb{E}[\|k(G', G')\|_{2;2}] = 4(\mathbb{E}[\|k(G, G)\|_{2;2}])^2,
\end{aligned}
$$

which implies

$$\mathbb{E}[\langle \Delta, k(G, G')\Delta'\rangle_{\mathbb{R}^m}^2] < \infty$$

since by assumption $\|k\|_{\mu_G, 2;2} < \infty$. $\qquad\square$

Since the unbiased estimator $\widehat{\mathrm{SKCE}}_{\mathrm{uq}}$ is a U-statistic, we know that the random variable $\sqrt{n}(\widehat{\mathrm{SKCE}}_{\mathrm{uq}} - \mathrm{SKCE})$ is asymptotically normally distributed under certain conditions.

**Theorem G.1.** *Let $k \colon \Delta^m \times \Delta^m \to \mathcal{L}(\mathbb{R}^m)$ be a kernel, and assume that $k(\cdot, t)u$ is measurable for all $t \in \Delta^m$ and $u \in \mathbb{R}^m$, and $\|k\|_{\mu_G, 1} < \infty$.*

*If $\mathrm{Var}[\langle \Delta, k(G, G')\Delta'\rangle_{\mathbb{R}^m}] < \infty$, then*

$$\sqrt{n}\left(\widehat{\mathrm{SKCE}}_{\mathrm{uq}}[k, g, \mathcal{D}] - \mathrm{SKCE}[k, g]\right) \xrightarrow{d} \mathcal{N}(0, 4\zeta_1),$$

*where*

$$\zeta_1 \coloneqq \mathbb{E}[\langle \Delta, k(G, G')\Delta'\rangle_{\mathbb{R}^m}\langle \Delta, k(G, G'')\Delta''\rangle_{\mathbb{R}^m}] - \mathrm{SKCE}^2[k, g],$$

*where $G'$ and $G''$ are independent copies of $G$ and $\Delta' \coloneqq \delta(G')$ and $\Delta'' \coloneqq \delta(G'')$.*

*Proof.* The statement follows immediately from van der Vaart (Theorem 12.3 1998). $\qquad\square$

If model $g$ is strongly calibrated, then $\zeta_1 = 0$, and hence $\widehat{\mathrm{SKCE}}_{\mathrm{uq}}$ is a so-called degenerate U-statistic (see, e.g., Section 12.3 van der Vaart, 1998).

**Theorem G.2.** *Let $k \colon \Delta^m \times \Delta^m \to \mathcal{L}(\mathbb{R}^m)$ be a kernel, and assume that $k(\cdot, t)u$ is measurable for all $t \in \Delta^m$ and $u \in \mathbb{R}^m$, and $\|k\|_{\mu_G, 2} < \infty$.*

*If $g$ is strongly calibrated, then*

$$n\,\widehat{\mathrm{SKCE}}_{\mathrm{uq}}[k, g, \mathcal{D}] \xrightarrow{d} \sum_{i=1}^{\infty} \lambda_i(Z_i^2 - 1),$$

*where $Z_i$ are independent standard normally distributed random variables and $\lambda_i$ with $\sum_{i=1}^{\infty} \lambda_i^2 < \infty$ are eigenvalues of the integral operator*

$$Kf(\xi, y) := \int \langle e_y - \xi, k(\xi, \xi')(e_{y'} - \xi')\rangle_{\mathbb{R}^m} f(\xi', y')\, \mu_{G \times Y}(\mathrm{d}(\xi', y'))$$

*on the space $L^2(\Delta^m \times \{1, \ldots, m\}, \mu_{G \times Y})$.*

*Proof.* From Lemma G.1 we know that

$$\mathrm{Var}[\langle \Delta, k(G, G')\Delta'\rangle_{\mathbb{R}^m}] < \infty.$$

Moreover, since $g$ is strongly calibrated, $\Delta = 0$ almost surely and by Theorem C.1 $\mathrm{KCE}[k, g] = 0$. Thus we obtain

$$\mathbb{E}[\langle \Delta, k(G, G')\Delta'\rangle_{\mathbb{R}^m}\langle \Delta, k(G, G'')\Delta''\rangle_{\mathbb{R}^m}] - \mathrm{SKCE}^2[k, g] = 0.$$

The statement follows from Serfling (Theorem in Section 5.5.2 1980). $\qquad\square$

As discussed by Gretton et al. (2012) in the case of two-sample tests, a natural idea is to find a threshold $c$ such that $\mathbb{P}[n\,\widehat{\mathrm{SKCE}}_{\mathrm{uq}}[k, g, \mathcal{D}] > c \mid H_0] \le \alpha$, where $H_0$ is the null hypothesis that the model is strongly calibrated. The desired quantile can be estimated by fitting Pearson curves to the empirical distribution by moment matching (Gretton et al., 2012), or alternatively by bootstrapping (Arcones and Giné, 1992), both computed and performed under the assumption that the model is strongly calibrated.

If model $g$ is strongly calibrated we know $\mathbb{E}[\widehat{\mathrm{SKCE}}_{\mathrm{uq}}[k, g, \mathcal{D}]] = 0$. Moreover, it follows from Hoeffding (p. 299 1948) that

$$\mathbb{E}\left[\widehat{\mathrm{SKCE}}_{\mathrm{uq}}^{2}[k, g, \mathcal{D}]\right] = \frac{2}{n(n-1)}\mathbb{E}\left[(\langle e_Y - g(X), k(g(X), g(X'))(e_{Y'} - g(X'))\rangle_{\mathbb{R}^m})^2\right],$$

where $(X', Y')$ is an independent copy of $(X, Y)$. By some tedious calculations we can retrieve higher-order moments as well. If model $g$ is strongly calibrated, we know from Serfling (1980, Lemma B, Section 5.2.2) that for $r \ge 2$

$$\mathbb{E}\left[\widehat{\mathrm{SKCE}}_{\mathrm{uq}}^{r}[k, g, \mathcal{D}]\right] = O(n^{-r})$$

as the number of samples $n$ goes to infinity, provided that

$$\mathbb{E}\left[|\langle e_Y - g(X), k(g(X), g(X'))(e_{Y'} - g(X'))\rangle_{\mathbb{R}^m}|^r\right] < \infty.$$

Alternatively, as discussed by Arcones and Giné (Section 5 1992), we can estimate $c$ by using quantiles of the bootstrap statistic

$$T = 2n^{-1}\sum_{1 \le i < j \le n}\left[h((X_{*,i}, Y_{*,i}), (X_{*,j}, Y_{*,j})) - n^{-1}\sum_{k=1}^{n}h((X_{*,i}, Y_{*,i}), (X_k, Y_k))\right.$$

$$\left. - n^{-1}\sum_{k=1}^{n}h((X_k, Y_k), (X_{*,j}, Y_{*,j})) + n^{-2}\sum_{k,l=1}^{n}h((X_k, Y_k), (X_l, Y_l))\right],$$

where

$$h((x, y), (x', y')) := \langle \delta(x, y), k(g(x), g(x'))\delta(x', y')\rangle_{\mathbb{R}^m}$$

and $(X_{*,1}, Y_{*,1}), \ldots, (X_{*,n}, Y_{*,n})$ are sampled with replacement from the data set $\mathcal{D}$. Then asymptotically

$$\mathbb{P}\left[ n \widehat{\mathrm{SKCE}}_{\mathrm{uq}}[k, g, \mathcal{D}] > c \,|\, H_0 \right] \approx \mathbb{P}[T > c \,|\, \mathcal{D}].$$

For the linear estimator, the asymptotical behaviour follows from the central limit theorem (e.g., Theorem A in Section 1.9 Serfling, 1980).

**Corollary G.1.** *Let $k\colon \Delta^m \times \Delta^m \to \mathcal{L}(\mathbb{R}^m)$ be a kernel, and assume that $k(\cdot, t)u$ is measurable for all $t \in \Delta^m$ and $u \in \mathbb{R}^m$, and $\|k\|_{\mu_G, 1} < \infty$.*

*If $\sigma^2 := \mathrm{Var}[\langle \delta(X, Y), k(g(X), g(X'))\delta(X', Y')\rangle_{\mathbb{R}^m}] < \infty$, where $(X', Y')$ is an independent copy of $(X, Y)$, then*

$$\sqrt{\lfloor n/2 \rfloor} \left( \widehat{\mathrm{SKCE}}_{\mathrm{ul}}[k, g, \mathcal{D}] - \mathrm{SKCE}[k, g] \right) \xrightarrow{d} \mathcal{N}(0, \sigma^2).$$

As noted in the following statement, the variance $\sigma^2$ is finite if $t \mapsto \|k(t, t)\|$ is $L^2$-integrable with respect to measure $\mu_G$.

**Corollary G.2.** *Let $k\colon \Delta^m \times \Delta^m \to \mathcal{L}(\mathbb{R}^m)$ be a kernel, and assume that $k(\cdot, t)u$ is measurable for all $t \in \Delta^m$ and $u \in \mathbb{R}^m$, and $\|k\|_{\mu_G, 2} < \infty$.*

*Then $\sigma^2 := \mathrm{Var}[\langle \delta(X, Y), k(G, G')\delta(X', Y')\rangle_{\mathbb{R}^m}] < \infty$, where $(X', Y')$ is an independent copy of $(X, Y)$ with $G' = g(X')$, and*

$$\sqrt{\lfloor n/2 \rfloor} \left( \widehat{\mathrm{SKCE}}_{\mathrm{ul}}[k, g, \mathcal{D}] - \mathrm{SKCE}[k, g] \right) \xrightarrow{d} \mathcal{N}(0, \sigma^2).$$

*Proof.* The statement follows from Corollary G.1. $\qquad\square$

The weak convergence of $\widehat{\mathrm{SKCE}}_{\mathrm{ul}}$ yields the following asymptotic test.

**Corollary G.3.** *Let the assumptions of Corollary G.1 be satisfied.*

*A one-sided statistical test with test statistic $\widehat{\mathrm{SKCE}}_{\mathrm{ul}}[k, g, \mathcal{D}]$ and asymptotic significance level $\alpha$ has the acceptance region*

$$\sqrt{\lfloor n/2 \rfloor} \, \widehat{\mathrm{SKCE}}_{\mathrm{ul}}[k, g, \mathcal{D}] < z_{1-\alpha}\hat{\sigma},$$

*where $z_{1-\alpha}$ is the $(1-\alpha)$-quantile of the standard normal distribution and $\hat{\sigma}$ is a consistent estimator of the standard deviation of $\langle \delta(X, Y), k(g(X), g(X'))\delta(X', Y')\rangle_{\mathbb{R}^m}$.*

## H  Distribution-free bounds

First we prove a helpful bound.

**Lemma H.1.** *Let $k\colon \Delta^m \times \Delta^m \to \mathcal{L}(\mathbb{R}^m)$ be a kernel, and assume that $K_{p;q} := \sup_{s,t \in \Delta^m} \|k(s, t)\|_{p;q} < \infty$ for some $1 \le p, q \le \infty$. Then*

$$\sup_{x,x' \in \mathcal{X}, y, y' \in \{1, \ldots, m\}} |\langle \delta(x, y), k(g(x), g(x'))\delta(x', y')\rangle_{\mathbb{R}^m}| \le 2^{1 + 1/p - 1/q} K_{p;q} =: B_{p;q}.$$

*Proof.* By Hölder's inequality and the definition of the operator norm for all $s, t \in \Delta^m$ and $u, v \in \mathbb{R}^m$

$$|\langle u, k(s, t)v\rangle_{\mathbb{R}^m}| \le \|u\|_{q'}\|k(s, t)v\|_q \le \|u\|_{q'}\|v\|_p\|k(s, t)\|_{p;q} \le K_{p;q}\|u\|_{q'}\|v\|_p.$$

The result follows from the fact that $\max_{s,t \in \Delta^m} \|s - t\|_p = 2^{1/p}$ and $\max_{s,t \in \Delta^m} \|s - t\|_{q'} = 2^{1/q'} = 2^{1-1/q}$. $\qquad\square$

Unfortunately, the tightness of the bound in Lemma H.1 depends on the choice of $p$ and $q$, as the following example shows.

**Example H.1**

Let $k = \phi \mathbf{I}_m$, where $\phi \colon \Delta^m \times \Delta^m \to \mathbb{R}$ is a scalar-valued kernel and $\mathbf{I}_m \in \mathbb{R}^{m \times m}$ is the identity matrix. Assume that $\Phi := \sup_{s,t \in \Delta^m} |\phi(s,t)| < \infty$. One can show that for all $s, t \in \Delta^m$

$$\|k(s,t)\|_{p;q} = \begin{cases} \phi(s,t) & \text{if } p \leq q, \\ m^{1/q - 1/p} \phi(s,t) & \text{if } p > q, \end{cases}$$

which implies that

$$K_{p;q} = \begin{cases} \Phi & \text{if } p \leq q, \\ m^{1/q - 1/p} \Phi & \text{if } p > q. \end{cases}$$

Thus the bound $B_{p;q}$ in Lemma H.1 is

$$B_{p;q} = \begin{cases} 2^{1+1/p-1/q} \Phi & \text{if } p \leq q, \\ 2^{1+1/p-1/q} m^{1/q-1/p} \Phi & \text{if } p > q, \end{cases}$$

which attains its smallest value $\min_{1 \leq p,q \leq \infty} B_{p;q} = 2\Phi$ if and only if $p = q$ or $m = 2$ and $p > q$. Thus for any other choice of $p$ and $q$ Lemma H.1 provides a non-optimal bound.

**Theorem H.1.** *Let $k \colon \Delta^m \times \Delta^m \to \mathcal{L}(\mathbb{R}^m)$ be a kernel, and assume that $k(\cdot, t)u$ is measurable for all $t \in \Delta^m$ and $u \in \mathbb{R}^m$, and $K_{p;q} := \sup_{s,t \in \Delta^m} \|k(s,t)\|_{p;q} < \infty$ for some $1 \leq p, q \leq \infty$. Then for all $\epsilon > 0$*

$$\mathbb{P}\left[ \left| \widehat{\mathrm{KCE}}[k,g,\mathcal{D}] - \mathrm{KCE}[k,g] \right| \geq 2(B_{p;q}/n)^{1/2} + \epsilon \right] \leq \exp\left( -\frac{\epsilon^2 n}{2B_{p;q}} \right).$$

*Proof.* Let $\mathcal{F}$ be the unit ball in the RKHS $\mathcal{H} \subset (\mathbb{R}^m)^{\Delta^m}$ corresponding to kernel $k$. We consider the random variable

$$F := \left| \widehat{\mathrm{KCE}}[k,g,\mathcal{D}] - \mathrm{KCE}[k,g] \right|.$$

The randomness of $F$ is due to the randomness of the data points $(X_i, Y_i)$, and by Lemmas E.2 and F.1 we can rewrite $F$ as

$$F = n^{-1} \left| \left\| \sum_{i=1}^{n} k(\cdot, g(X_i)) \delta(X_i, Y_i) \right\|_{\mathcal{H}} - n \|\mu_g\|_{\mathcal{H}} \right| =: f((X_1, Y_1), \ldots, (X_n, Y_n)),$$

where $\mu_g$ is the embedding defined in Lemma E.1. The triangle inequality implies that for all $z_i = (x_i, y_i) \in \mathcal{X} \times \{1, \ldots, m\}$

$$
\begin{aligned}
f(z_1, \ldots, z_n) &= n^{-1} \left| \left\| \sum_{i=1}^{n} k(\cdot, g(x_i)) \delta(x_i, y_i) \right\|_{\mathcal{H}} - n\|\mu_g\|_{\mathcal{H}} \right| \\
&\leq n^{-1} \left\| \sum_{i=1}^{n} \left( k(\cdot, g(x_i)) \delta(x_i, y_i) - \mu_g \right) \right\|_{\mathcal{H}} =: h(z_1, \ldots, z_n),
\end{aligned}
\tag{H.1}
$$

where $h \colon (\mathcal{X} \times \{1, \ldots, m\})^n \to \mathbb{R}$ is measurable and hence induces a random variable $H := h((X_1, Y_1), \ldots, (X_n, Y_n))$.

By the reproducing property and Lemma H.1, for all $x, x' \in \mathcal{X}$ and $y, y' \in \{1, \ldots, m\}$ we have

$$
\begin{aligned}
\|k(\cdot, g(x)) \delta(x,y) - k(\cdot, g(x')) \delta(x',y')\|_{\mathcal{H}}^2 &= \langle \delta(x,y), k(g(x), g(x)) \delta(x,y) \rangle_{\mathbb{R}^m} \\
&\quad - \langle \delta(x,y), k(g(x), g(x')) \delta(x',y') \rangle_{\mathbb{R}^m} \\
&\quad - \langle \delta(x',y'), k(g(x'), g(x)) \delta(x,y) \rangle_{\mathbb{R}^m} \\
&\quad + \langle \delta(x',y'), k(g(x'), g(x')) \delta(x',y') \rangle_{\mathbb{R}^m} \\
&\leq 4 B_{p;q}.
\end{aligned}
$$

Thus for all $i \in \{1, \ldots, m\}$ the triangle inequality implies

$$\sup_{z, z', z_j (j \neq i)} |h(z_1, \ldots, z_{i-1}, z, z_{i+1}, \ldots, z_n) - h(z_1, \ldots, z_{i-1}, z', z_{i+1}, \ldots, z_n)|$$

$$\leq \sup_{x, x, y, y'} n^{-1} \|k(\cdot, g(x)) \delta(x,y) - k(\cdot, g(x')) \delta(x',y')\|_{\mathcal{H}} \leq \frac{2 B_{p;q}^{1/2}}{n}.$$

Hence we can apply McDiarmid's inequality to the random variable $H$, which yields for all $\epsilon > 0$

$$\mathbb{P}\left[H \geq \mathbb{E}[H] + \epsilon\right] \leq \exp\left(-\frac{\epsilon^2 n}{2B_{p;q}}\right). \tag{H.2}$$

In the final parts of the proof we bound the expectation $\mathbb{E}[H]$. By Lemmas E.1 and F.1, we know that

$$\begin{aligned}
H &= h((X_1, Y_1), \ldots, (X_n, Y_n)) \\
&= \sup_{f \in \mathcal{F}} n^{-1} \left| \sum_{i=1}^{n} \left( \langle \delta(X_i, Y_i), f(g(X_i)) \rangle_{\mathbb{R}^m} - \mathbb{E}\left[ \langle \delta(X, Y), f(g(X)) \rangle_{\mathbb{R}^m} \right] \right) \right| \\
&= \sup_{f \in \mathcal{F}_0} n^{-1} \left| \sum_{i=1}^{n} f(X_i, Y_i) - \mathbb{E}[f(X, Y)] \right|,
\end{aligned}$$

where $\mathcal{F}_0 := \{f \colon \mathcal{X} \times \{1, \ldots, m\} \to \mathbb{R}, (x, y) \mapsto \langle \delta(x, y), \tilde{f}(g(x)) \rangle_{\mathbb{R}^m} \colon \tilde{f} \in \mathcal{F}\}$ is a class of measurable functions. As Gretton et al. (2012), we make use of symmetrization ideas (van der Vaart and Wellner, 1996, p. 108). From van der Vaart and Wellner (1996, Lemma 2.3.1) it follows that

$$\mathbb{E}[H] = \mathbb{E}\left[ \sup_{f \in \mathcal{F}_0} n^{-1} \left| \sum_{i=1}^{n} f(X_i, Y_i) - \mathbb{E}[f(X, Y)] \right| \right] \leq 2\,\mathbb{E}\left[ \sup_{f \in \mathcal{F}_0} \left| n^{-1} \sum_{i=1}^{n} \epsilon_i f(X_i, Y_i) \right| \right],$$

where $\epsilon_1, \ldots, \epsilon_n$ are independent Rademacher random variables. Similar to Bartlett and Mendelson (2002, Lemma 22), we obtain

$$\begin{aligned}
\mathbb{E}[H] &\leq 2n^{-1}\,\mathbb{E}\left[ \sup_{f \in \mathcal{F}} \left| \sum_{i=1}^{n} \epsilon_i \langle \delta(X_i, Y_i), f(g(X_i)) \rangle_{\mathbb{R}^m} \right| \right] \\
&= 2n^{-1}\,\mathbb{E}\left[ \sup_{f \in \mathcal{F}} \left| \left\langle \sum_{i=1}^{n} \epsilon_i k(\cdot, g(X_i)) \delta(X_i, Y_i), f \right\rangle_{\mathcal{H}} \right| \right] \\
&= 2n^{-1}\,\mathbb{E}\left[ \left\| \sum_{i=1}^{n} \epsilon_i k(\cdot, g(X_i)) \delta(X_i, Y_i) \right\|_{\mathcal{H}} \right] \\
&= 2n^{-1}\,\mathbb{E}\left[ \left( \sum_{i,j=1}^{n} \epsilon_i \epsilon_j \langle k(\cdot, g(X_i)) \delta(X_i, Y_i), k(\cdot, g(X_j)) \delta(X_j, Y_j) \rangle_{\mathcal{H}} \right)^{1/2} \right].
\end{aligned}$$

By Jensen's inequality we get

$$\begin{aligned}
\mathbb{E}[H] &\leq 2n^{-1} \left( \sum_{i,j=1}^{n} \mathbb{E}\left[ \epsilon_i \epsilon_j \langle k(\cdot, g(X_i)) \delta(X_i, Y_i), k(\cdot, g(X_j)) \delta(X_j, Y_j) \rangle_{\mathcal{H}} \right] \right)^{1/2} \\
&= 2n^{-1/2} \left( \mathbb{E}\left[ \langle k(\cdot, g(X)) \delta(X, Y), k(\cdot, g(X)) \delta(X, Y) \rangle_{\mathcal{H}} \right] \right)^{1/2} \\
&\leq 2(B_{p;q}/n)^{1/2}.
\end{aligned} \tag{H.3}$$

All in all, from Eqs. (H.1) to (H.3) we obtain for all $\epsilon > 0$

$$\begin{aligned}
\mathbb{P}\left[ \left| \widehat{\mathrm{KCE}}[k, g, \mathcal{D}] - \mathrm{KCE}[k, g] \right| \geq 2(B_{p;q}/n)^{1/2} + \epsilon \right] &= \mathbb{P}[F \geq 2(B_{p;q}/n)^{1/2} + \epsilon] \\
&\leq \mathbb{P}[H \geq 2(B_{p;q}/n)^{1/2} + \epsilon] \\
&\leq \mathbb{P}[H \geq \mathbb{E}[H] + \epsilon] \\
&\leq \exp\left( -\frac{\epsilon^2 n}{2B_{p;q}} \right),
\end{aligned}$$

which concludes our proof. $\qquad \square$

If model $g$ is calibrated in the strong sense, we can improve the bound.

**Theorem H.2.** *Let $k\colon \Delta^m \times \Delta^m \to \mathcal{L}(\mathbb{R}^m)$ be a kernel, and assume that $k(\cdot, t)u$ is measurable for all $t \in \Delta^m$ and $u \in \mathbb{R}^m$, and $K_{p;q} := \sup_{s,t \in \Delta^m} \|k(s,t)\|_{p;q} < \infty$ for some $1 \leq p, q \leq \infty$. Define*

$$B_1 := n^{-1/2} \left[ \mathbb{E}\left[ \langle \delta(X,Y), k(g(X), g(X))\delta(X,Y) \rangle_{\mathbb{R}^m} \right] \right]^{1/2}, \qquad and$$

$$B_2 := (B_{p;q}/n)^{1/2}.$$

*Then $B_1 \leq B_2$, and for all $\epsilon > 0$ and $i \in \{1, 2\}$*

$$\mathbb{P}\left[ \widehat{\mathrm{KCE}}[k, g, \mathcal{D}] \geq B_i + \epsilon \right] \leq \exp\left( -\frac{\epsilon^2 n}{2B_{p;q}} \right),$$

*if $g$ is calibrated in the strong sense.*

*Proof.* Let $\mathcal{F}$ be the unit ball in the RKHS $\mathcal{H} \subset (\mathbb{R}^m)^{\Delta^m}$ corresponding to kernel $k$. Lemma H.1 implies

$$
\begin{aligned}
B_1 &= n^{-1/2} \left[ \mathbb{E}\left[ \langle \delta(X,Y), k(g(X), g(X))\delta(X,Y) \rangle_{\mathbb{R}^m} \right] \right]^{1/2} \\
&\leq n^{-1/2} \left[ \mathbb{E}[B_{p;q}] \right]^{1/2} = (B_{p;q}/n)^{1/2} = B_2.
\end{aligned}
\tag{H.4}
$$

Let $H$ be defined as in the proof of Theorem H.1. Since $g$ is strongly calibrated, it follows from Theorem C.1 and Lemma E.2 that $\mu_g = 0$, and thus by Lemma F.1

$$H = n^{-1} \left\| \sum_{i=1}^n k(\cdot, g(x_i))\delta(x_i, y_i) \right\|_{\mathcal{H}} = \widehat{\mathrm{KCE}}[k, g, \mathcal{D}].$$

Thus Eq. (H.2) implies

$$\mathbb{P}\left[ \widehat{\mathrm{KCE}}[k, g, \mathcal{D}] \geq \mathbb{E}[\widehat{\mathrm{KCE}}[k, g, \mathcal{D}]] + \epsilon \right] \leq \exp\left( -\frac{\epsilon^2 n}{2B_{p;q}} \right). \tag{H.5}$$

Next we bound $\mathbb{E}[\widehat{\mathrm{KCE}}[k, g, \mathcal{D}]]$. From Lemma F.1 we get

$$\mathbb{E}[\widehat{\mathrm{KCE}}[k, g, \mathcal{D}]] = \frac{1}{n} \mathbb{E}\left[ \left( \sum_{i,j=1}^n \langle \delta(X_i, Y_i), k(g(X_i), g(X_j))\delta(X_j, Y_j) \rangle_{\mathbb{R}^m} \right)^{1/2} \right],$$

and hence by Jensen's inequality we obtain

$$
\begin{aligned}
\mathbb{E}[\widehat{\mathrm{KCE}}[k, g, \mathcal{D}]] &\leq \frac{1}{n} \left( \mathbb{E}\left[ \sum_{i,j=1}^n \langle \delta(X_i, Y_i), k(g(X_i), g(X_j))\delta(X_j, Y_j) \rangle_{\mathbb{R}^m} \right] \right)^{1/2} \\
&= \frac{1}{n} \Big( n\, \mathbb{E}\left[ \langle \delta(X,Y), k(g(X), g(X))\delta(X,Y) \rangle_{\mathbb{R}^m} \right] \\
&\qquad + n(n-1)\, \mathbb{E}\left[ \langle \delta(X,Y), k(g(X), g(X'))\delta(X', Y') \rangle_{\mathbb{R}^m} \right] \Big)^{1/2},
\end{aligned}
$$

where $(X', Y')$ denotes an independent copy of $(X, Y)$. From Lemma E.2 it follows that

$$\mathbb{E}[\widehat{\mathrm{KCE}}[k, g, \mathcal{D}]] \leq \left( \frac{1}{n} \mathbb{E}\left[ \langle \delta(X,Y), k(g(X), g(X))\delta(X,Y) \rangle_{\mathbb{R}^m} \right] + \left( 1 - \frac{1}{n} \right) \mathrm{SKCE}[k, g] \right)^{1/2}.$$

If model $g$ is calibrated in the strong sense, we know from Theorem C.1 that $\mathrm{SKCE}[k, g] = 0$. Thus we obtain

$$\mathbb{E}[\widehat{\mathrm{KCE}}[k, g, \mathcal{D}]] \leq B_1. \tag{H.6}$$

All in all, from Eqs. (H.4) to (H.6) it follows that for all $\epsilon > 0$ and $i \in \{1, 2\}$

$$\mathbb{P}\left[\widehat{\mathrm{KCE}}[k, g, \mathcal{D}] \geq B_i + \epsilon\right] \leq \mathbb{P}\left[\widehat{\mathrm{KCE}}[k, g, \mathcal{D}] \geq B_1 + \epsilon\right]$$
$$\leq \mathbb{P}\left[\widehat{\mathrm{KCE}}[k, g, \mathcal{D}] \geq \mathbb{E}[\widehat{\mathrm{KCE}}[k, g, \mathcal{D}] + \epsilon\right]$$
$$\leq \exp\left(-\frac{-\epsilon^2 n}{2 B_{p;q}}\right),$$

if $g$ is calibrated in the strong sense. $\qquad \square$

Thus we obtain the following distribution-free hypothesis test.

**Corollary H.1.** *Let the assumptions of Theorem H.2 be satisfied.*

*A statistical test with test statistic $\widehat{\mathrm{KCE}}[k, g, \mathcal{D}]$ and significance level $\alpha$ for the null hypothesis of model $g$ being calibrated in the strong sense has the acceptance region*

$$\widehat{\mathrm{KCE}}[k, g, \mathcal{D}] < (B_{p;q}/n)^{1/2}(1 + \sqrt{-2 \log \alpha}).$$

A distribution-free bound for the deviation of the unbiased estimator can be obtained from the theory of U-statistics.

**Theorem H.3.** *Let $k \colon \Delta^m \times \Delta^m \to \mathcal{L}(\mathbb{R}^m)$ be a kernel, and assume that $k(\cdot, t)u$ is measurable for all $t \in \Delta^m$ and $u \in \mathbb{R}^m$, and $K_{p;q} := \sup_{s,t \in \Delta^m} \|k(s, t)\|_{p;q}$ for some $1 \leq p, q \leq \infty$. Then for all $t > 0$*

$$\mathbb{P}\left[\widehat{\mathrm{SKCE}}_{\mathrm{uq}}[k, g, \mathcal{D}] - \mathrm{SKCE}[k, g] \geq t\right] \leq \exp\left(-\frac{\lfloor n/2 \rfloor t^2}{2 B_{p;q}^2}\right).$$

*The same bound holds for $\mathbb{P}\left[\widehat{\mathrm{SKCE}}_{\mathrm{uq}}[k, g, \mathcal{D}] - \mathrm{SKCE}[k, g] \leq -t\right]$.*

*Proof.* By Lemma F.2, $\mathbb{E}[\widehat{\mathrm{SKCE}}_{\mathrm{uq}}[k, g, \mathcal{D}]] = \mathrm{SKCE}[k, g]$. Moreover, by Lemma H.1 we know that
$$\sup_{x,x' \in \mathcal{X}, y, y' \in \{1, \ldots, m\}} |\langle \delta(x, y), k(g(x), g(x'))\delta(x', y')\rangle_{\mathbb{R}^m}| \leq B_{p;q}.$$
Thus the result follows from the bound on U-statistics by Hoeffding (1963, p. 25). $\qquad \square$

We can derive a hypothesis test using the unbiased estimator.

**Corollary H.2.** *Let the assumptions of Theorem H.3 be satisfied.*

*A one-sided statistical test with test statistic $\widehat{\mathrm{SKCE}}_{\mathrm{uq}}[k, g, \mathcal{D}]$ and significance level $\alpha$ for the null hypothesis of model $g$ being calibrated in the strong sense has the acceptance region*

$$\widehat{\mathrm{SKCE}}_{\mathrm{uq}}[k, g, \mathcal{D}] < \frac{B_{p;q}}{\sqrt{\lfloor n/2 \rfloor}}\sqrt{-2 \log \alpha}.$$

Analogously we can obtain a bound for the linear estimator.

**Theorem H.4.** *Let $k \colon \Delta^m \times \Delta^m \to \mathcal{L}(\mathbb{R}^m)$ be a kernel, and assume that $k(\cdot, t)u$ is measurable for all $t \in \Delta^m$ and $u \in \mathbb{R}^m$, and $K_{p;q} := \sup_{s,t \in \Delta^m} \|k(s, t)\|_{p;q}$ for some $1 \leq p, q \leq \infty$. Then for all $t > 0$*

$$\mathbb{P}\left[\widehat{\mathrm{SKCE}}_{\mathrm{ul}}[k, g, \mathcal{D}] - \mathrm{SKCE}[k, g] \geq t\right] \leq \exp\left(-\frac{\lfloor n/2 \rfloor t^2}{2 B_{p;q}^2}\right).$$

*The same bound holds for $\mathbb{P}\left[\widehat{\mathrm{SKCE}}_{\mathrm{ul}}[k, g, \mathcal{D}] - \mathrm{SKCE}[k, g] \leq -t\right]$.*

*Proof.* By Lemma F.3, $\mathbb{E}[\widehat{\mathrm{SKCE}}_{\mathrm{ul}}[k, g, \mathcal{D}]] = \mathrm{SKCE}[k, g]$. Moreover, by Lemma H.1 we know that
$$\sup_{x,x' \in \mathcal{X}, y, y' \in \{1, \ldots, m\}} |\langle \delta(x, y), k(g(x), g(x'))\delta(x', y')\rangle_{\mathbb{R}^m}| \leq B_{p;q}.$$
Thus by Hoeffding's inequality (Hoeffding, 1963, Theorem 2) for all $t > 0$

$$\mathbb{P}\left[\widehat{\mathrm{SKCE}}_{\mathrm{ul}}[k, g, \mathcal{D}] - \mathrm{SKCE}[k, g] \geq t\right] \leq \exp\left(-\frac{\lfloor n/2 \rfloor t^2}{2 B_{p;q}^2}\right). \qquad \square$$

Obviously this results yields another distribution-free hypothesis test.

**Corollary H.3.** *Let the assumptions of Theorem H.4 be satisfied.*

*A one-sided statistical test with test statistic $\widehat{\mathrm{SKCE}}_{\mathrm{ul}}[k, g, \mathcal{D}]$ and significance level $\alpha$ for the null hypothesis of model $g$ being calibrated in the strong sense has the acceptance region*

$$\widehat{\mathrm{SKCE}}_{\mathrm{ul}}[k, g, \mathcal{D}] < \frac{B_{p;q}}{\sqrt{\lfloor n/2 \rfloor}} \sqrt{-2 \log \alpha}.$$

# I Comparisons

## I.1 Expected calibration error and maximum calibration error

For certain spaces of bounded functions the calibration error CE turns out to be a form of the ECE. In particular, the ECE with respect to the cityblock distance, the total variation distance, and the squared Euclidean distance are special cases of CE. Choosing $p = 1$ in the following statement corresponds to the special case of the MCE.

**Lemma I.1** (ECE **and** MCE **as special cases**). *Let $1 \leq p \leq \infty$ with Hölder conjugate $p'$. If $\mathcal{F} = K^p(\Delta^m, \mu_G; \mathbb{R}^m)$, then $\mathrm{CE}[\mathcal{F}, g] = \|\delta\|_{\mu_G, p'}$.*

*Proof.* Note that $\mathcal{F}$ is well defined since $\mathcal{F} \subset L^1(\Delta^m, \mu_G; \mathbb{R}^m)$.

The statement follows from the extremal case of Hölder's inequality. More explicitly, let $\nu$ denote the counting measure on $\{1, \dots, m\}$. Since both $\mu_G$ and $\nu$ are $\sigma$-finite measures, the product measure $\mu_G \otimes \nu$ on the product space $B := \Delta^m \times \{1, \dots, m\}$ is uniquely defined and $\sigma$-finite. Define $\tilde{\delta}(t, k) := \delta_k(t)$ for all $(t, k) \in B$. Then we can rewrite

$$\begin{aligned}
\mathrm{CE}[\mathcal{F}, g] &= \sup_{f \in K^p(\Delta^m, \mu_G; \mathbb{R}^m)} \int_{\Delta^m} \langle \delta(x), f(x) \rangle_{\mathbb{R}^m} \, \mu_G(\mathrm{d}x) \\
&= \sup_{f \in K^p(B, \mu_G \otimes \nu; \mathbb{R}^m)} \int_B |\tilde{\delta}(x, k) f(x, k)| \, (\mu_G \times \nu)(\mathrm{d}(x, k)) \\
&= \|\tilde{\delta}\|_{\mu_G \otimes \nu, p'} = \|\delta\|_{\mu_G, p'},
\end{aligned}$$

to make the reasoning more apparent. Since $\mu_G \otimes \nu$ is $\sigma$-finite the statement holds even for $p = 1$. $\square$

## I.2 Maximum mean calibration error

The so-called "correctness score" (Kumar et al. (2018)) $c(x, y)$ of an input $x$ and a class $y$ is defined as $c(x, y) = \mathbb{1}_{\{\arg\max_{y'} g_{y'}(x)\}}(y)$. It is 1 if class $y$ is equal to the class that is most likely for input $x$ according to model $g$, and 0 otherwise. Let $k \colon [0, 1] \times [0, 1] \to \mathbb{R}$ be a scalar-valued kernel. Then the maximum mean calibration error $\mathrm{MMCE}[k, g]$ of a model $g$ with respect to kernel $k$ is defined[6] as

$$\mathrm{MMCE}[k, g]$$
$$= \left( \mathbb{E}[(c(X, Y) - g_{\max}(X))(c(X', Y') - g_{\max}(X'))k(g_{\max}(X), g_{\max}(X'))] \right)^{1/2},$$

where $(X', Y')$ is an independent copy of $(X, Y)$.

Example I.1 shows that the KCE allows exactly the same analysis of the common notion of calibration as the MMCE proposed by Kumar et al. (2018) by applying it to a model that is reduced to the most confident predictions.

**Example I.1 (MMCE as special case)**
Reduce model $g$ to its most confident predictions by defining a new model $\tilde{g}$ with $\tilde{g}(x) := (g_{\max}(x), 1 - g_{\max}(x))$. The predictions $\tilde{g}(x)$ of this new model can be viewed as a model of

the conditional probabilities $(\mathbb{P}[\tilde{Y} = 1 \mid X = x], \mathbb{P}[\tilde{Y} = 2 \mid X = x])$ in a classification problem with inputs $X$ and classes $\tilde{Y} := 2 - c(X, Y)$.[7]

Let $k \colon [0, 1] \times [0, 1] \to \mathbb{R}$ be a scalar-valued kernel. Define a matrix-valued function $\tilde{k} \colon \Delta^2 \times \Delta^2 \to \mathbb{R}^{2 \times 2}$ by

$$\tilde{k}((p_1, p_2), (q_1, q_2)) = \frac{k(p_1, q_1)}{2} \mathbf{I}_2.$$

Then by Caponnetto et al. (2008, Example 1 and Theorem 14) $\tilde{k}$ is a matrix-valued kernel and, if it is continuous, it is universal if and only if $k$ is universal. By construction $e_{\tilde{Y}} - \tilde{g}(X) = (c(X, Y) - g_{\max}(X))(1, -1)$, and hence

$$
\begin{aligned}
\mathrm{SKCE}[\tilde{k}, \tilde{g}] &= \mathbb{E}[(e_{\tilde{Y}} - \tilde{g}(X))^{\intercal} \tilde{k}(\tilde{g}(X), \tilde{g}(X'))(e_{\tilde{Y}} - \tilde{g}(X'))] \\
&= \mathbb{E}[(c(X, Y) - g_{\max}(X))(c(X', Y') - g_{\max}(X'))k(g_{\max}(X), g_{\max}(X'))] \\
&= \mathrm{MMCE}^2[k, g],
\end{aligned}
$$

where $(X', \tilde{Y}')$ and $(X', Y')$ are independent copies of $(X, \tilde{Y})$ and $(X, Y)$, respectively.

# J  Experiments

The Julia implementation for all experiments is available online at `https://github.com/devmotion/CalibrationPaper`. The code is written and documented with the literate programming tool Weave.jl Pastell (2017) and exported to HTML files that include results and figures.

## J.1  Calibration errors

In our experiments we evaluate the proposed estimators of the SKCE and compare them with two estimators of the ECE.

### J.1.1  Expected calibration error

As commonly done (Bröcker and Smith, 2007; Guo et al., 2017; Vaicenavicius et al., 2019), we study the ECE with respect to the total variation distance.

The standard histogram-regression estimator of the ECE is based on a partitioning of the probability simplex (Guo et al., 2017; Vaicenavicius et al., 2019). In our experiments we use two different partitioning schemes. The first scheme is the commonly used partitioning into bins of uniform size, based on splitting the predictions of each class into 10 bins. The other partitioning is data-dependent: the data set is split iteratively along the median of the class predictions with the highest variance as long as the number of samples in a bin is at least 10.

### J.1.2  Kernel calibration error

We consider the matrix-valued kernel $k(x, y) = \exp\left(-\|x - y\|/\nu\right)\mathbf{I}_m$ with kernel bandwidth $\nu > 0$. Analogously to the ECE, we take the total variation distance as distance measure. Moreover, we choose the bandwidth adaptively with the so-called median heuristic. The median heuristic is a common heuristic that proposes to set the bandwidth to the median of the pairwise distances of samples in a, not necessarily separate, validation data set (see, e.g., Gretton et al., 2012).

## J.2  Generative models

Since the considered calibration errors depend only on the predictions and labels, we specify generative models of labeled predictions $(g(X), Y)$ without considering $X$. Instead we only specify the distribution of the predictions $g(X)$ and the conditional distribution of $Y$ given $g(X) = g(x)$. This setup allows us to design calibrated and uncalibrated models in a straightforward way, which enables clean numerical evaluations with known calibration errors.

We study the generative model

$$
\begin{aligned}
g(X) &\sim \mathrm{Dir}(\alpha), \\
Z &\sim \mathrm{Ber}(\pi), \\
Y \mid Z = 1, g(X) = \gamma &\sim \mathrm{Cat}(\beta), \\
Y \mid Z = 0, g(X) = \gamma &\sim \mathrm{Cat}(\gamma),
\end{aligned}
$$

with parameters $\alpha \in \mathbb{R}^m_{>0}$, $\beta \in \Delta^m$, and $\pi \in [0, 1]$. The model is calibrated if and only if $\pi = 0$, since for all labels $y \in \{1, \ldots, m\}$ we obtain

$$
\mathbb{P}[Y = y \mid g(X)] = \pi\beta_y + (1 - \pi)g_y(X),
$$

and hence $\Delta = \pi(\beta - g(X)) = 0$ almost surely if and only if $\pi = 0$.

By setting $\alpha = (1, \ldots, 1)$ we can model uniformly distributed predictions, and by decreasing the magnitude of $\alpha$ we can push the predictions towards the edges of the probability simplex, mimicking the predictions of a trained model (cf., e.g., Vaicenavicius et al., 2019).

### J.2.1 Theoretical expected calibration error

For the considered model, the ECE with respect to the total variation distance is

$$\mathrm{ECE}[\|.\|_{\mathrm{TV}}, g] = \mathbb{E}[\|\Delta\|_{\mathrm{TV}}] = \pi\, \mathbb{E}[\|\beta - g(X)\|_{\mathrm{TV}}] = \pi/2 \sum_{i=1}^{m} \mathbb{E}[|\beta_i - g_i(X)|]$$

$$= \frac{\pi}{2} \sum_{i=1}^{m} \left( \left( \frac{\alpha_i}{\alpha_0} - \beta_i \right) \left( 1 - \frac{2B(\beta_i; \alpha_i, \alpha_0 - \alpha_i)}{B(\alpha_i, \alpha_0 - \alpha_i)} \right) + \frac{2\beta_i^{\alpha_i}(1 - \beta_i)^{\alpha_0 - \alpha_i}}{\alpha_0 B(\alpha_i, \alpha_0 - \alpha_i)} \right),$$

where $\alpha_0 := \sum_{i=1}^{m} \alpha_i$ and $B(x; a, b)$ denotes the incomplete Beta function $\int_0^x t^{a-1}(1-t)^{b-1}\, \mathrm{d}t$. By exploiting $\sum_{i=1}^{m} \beta_i = 1$, we get

$$\mathrm{ECE}[\|.\|_{\mathrm{TV}}, g] = \frac{\pi}{\alpha_0} \sum_{i=1}^{m} \frac{(\alpha_0 \beta_i - \alpha_i) B(\beta_i; \alpha_i, \alpha_0 - \alpha_i) + \beta_i^{\alpha_i}(1 - \beta_i)^{\alpha_0 - \alpha_i}}{B(\alpha_i, \alpha_0 - \alpha_i)}.$$

Let $I(x; a, b) := B(x; a, b)/B(a, b)$ denote the regularized incomplete Beta function. Due to the identity $x^a(1-x)^b/B(a, b) = a(I(x; a, b) - I(x; a+1, b))$, we obtain

$$\mathrm{ECE}[\|.\|_{\mathrm{TV}}, g] = \pi \sum_{i=1}^{m} \left( \beta_i I(\beta_i; \alpha_i, \alpha_0 - \alpha_i) - \frac{\alpha_i}{\alpha_0} I(\beta_i; \alpha_i + 1, \alpha_0 - \alpha_i) \right).$$

If $\alpha = (a, \ldots, a)$ for some $a > 0$, then

$$\mathrm{ECE}[\|.\|_{\mathrm{TV}}, g] = \pi \sum_{i=1}^{m} \left( \beta_i I(\beta_i; a, (m-1)a) - m^{-1} I(\beta_i; a+1, (m-1)a) \right).$$

If $\beta = e_j$ for some $j \in \{1, \ldots, m\}$ we get

$$\mathrm{ECE}[\|.\|_{\mathrm{TV}}, g] = \pi \left( I(1; a, (m-1)a) - m^{-1} I(1; a+1, (m-1)a) \right)$$

$$= \pi(1 - m^{-1}) = \frac{\pi(m-1)}{m},$$

whereas if $\beta = (1/m, \ldots, 1/m)$ we obtain

$$\mathrm{ECE}[\|.\|_{\mathrm{TV}}, g] = \pi \left( I(m^{-1}; a, (m-1)a) - I(m^{-1}; a+1, (m-1)a) \right)$$

$$= \pi \frac{m^{-a}(1 - m^{-1})^{(m-1)a}}{a B(a, (m-1)a)} = \frac{\pi}{a B(a, (m-1)a)} \left( \frac{(m-1)^{m-1}}{m^m} \right)^a.$$

We see that, as the number of classes goes to infinity, the ECE with respect to the total variation distance tends to $\pi$ and $\pi \exp(-a) a^{a-1}/\Gamma(a)$, respectively.

### J.2.2 Mean total variation distance

For the considered generative models, we can compute the mean total variation distance $\mathbb{E}[\|X - X'\|_{\mathrm{TV}}]$, which does not depend on the number of available samples (but, of course, is usually not available). If $X$ and $X'$ are i.i.d. according to $\mathrm{Dir}(\alpha)$ with parameter $\alpha \in \mathbb{R}_{>0}^m$, then their mean total variation distance is

$$\mathbb{E}[\|X - X'\|_{\mathrm{TV}}] = 1/2 \sum_{i=1}^{m} \mathbb{E}[|X_i - X_i'|]$$

$$= \sum_{i=1}^{m} \mathbb{E}[X_i - X_i' \mid X_i > X_i']$$

$$= \frac{2B(\alpha_0, \alpha_0)}{\alpha_0} \sum_{i=1}^{m} [B(\alpha_i, \alpha_i) B(\alpha_0 - \alpha_i, \alpha_0 - \alpha_i)]^{-1},$$

where $\alpha_0 := \sum_{i=1}^{m} \alpha_i$. We conduct additional experiments in which we set the kernel bandwidth to the mean total variation distance.

### J.2.3 Distribution of estimates: Additional figures

Figure 3: Distribution of $\widehat{\text{ECE}}$ with bins of uniform size, evaluated on $10^4$ data sets of 250 labeled predictions that are randomly sampled from generative models with $\alpha = (1, \ldots, 1)$ and $\beta = (1/m, \ldots, 1/m)$.

Figure 4: Distribution of $\widehat{\text{ECE}}$ with bins of uniform size, evaluated on $10^4$ data sets of 250 labeled predictions that are randomly sampled from generative models with $\alpha = (0.1, \ldots, 0.1)$ and $\beta = (1/m, \ldots, 1/m)$.

Figure 5: Distribution of $\widehat{\mathrm{ECE}}$ with bins of uniform size, evaluated on $10^4$ data sets of 250 labeled predictions that are randomly sampled from generative models with $\alpha = (1, \ldots, 1)$ and $\beta = (1, 0, \ldots, 0)$.

Figure 6: Distribution of $\widehat{\mathrm{ECE}}$ with bins of uniform size, evaluated on $10^4$ data sets of 250 labeled predictions that are randomly sampled from generative models with $\alpha = (0.1, \ldots, 0.1)$ and $\beta = (1, 0, \ldots, 0)$.

Figure 7: Distribution of $\widehat{\mathrm{ECE}}$ with data-dependent bins, evaluated on $10^4$ data sets of 250 labeled predictions that are randomly sampled from generative models with $\alpha = (1,\ldots,1)$ and $\beta = (1/m,\ldots,1/m)$.

Figure 8: Distribution of $\widehat{\mathrm{ECE}}$ with data-dependent bins, evaluated on $10^4$ data sets of 250 labeled predictions that are randomly sampled from generative models with $\alpha = (0.1,\ldots,0.1)$ and $\beta = (1/m,\ldots,1/m)$.

Figure 9: Distribution of $\widehat{\mathrm{ECE}}$ with data-dependent bins, evaluated on $10^4$ data sets of 250 labeled predictions that are randomly sampled from generative models with $\alpha = (1, \ldots, 1)$ and $\beta = (1, 0, \ldots, 0)$.

Figure 10: Distribution of $\widehat{\mathrm{ECE}}$ with data-dependent bins, evaluated on $10^4$ data sets of 250 labeled predictions that are randomly sampled from generative models with $\alpha = (0.1, \ldots, 0.1)$ and $\beta = (1, 0, \ldots, 0)$.

Figure 11: Distribution of $\widehat{\mathrm{SKCE}}_{\mathrm{b}}$ with the median heuristic, evaluated on $10^4$ data sets of 250 labeled predictions that are randomly sampled from generative models with $\alpha = (1, \ldots, 1)$ and $\beta = (1/m, \ldots, 1/m)$.

Figure 12: Distribution of $\widehat{\mathrm{SKCE}}_{\mathrm{b}}$ with the median heuristic, evaluated on $10^4$ data sets of 250 labeled predictions that are randomly sampled from generative models with $\alpha = (0.1, \ldots, 0.1)$ and $\beta = (1/m, \ldots, 1/m)$.

Figure 13: Distribution of $\widehat{\text{SKCE}}_{\text{b}}$ with the median heuristic, evaluated on $10^4$ data sets of 250 labeled predictions that are randomly sampled from generative models with $\alpha = (1, \ldots, 1)$ and $\beta = (1, 0, \ldots, 0)$.

Figure 14: Distribution of $\widehat{\text{SKCE}}_{\text{b}}$ with the median heuristic, evaluated on $10^4$ data sets of 250 labeled predictions that are randomly sampled from generative models with $\alpha = (0.1, \ldots, 0.1)$ and $\beta = (1, 0, \ldots, 0)$.

Figure 15: Distribution of $\widehat{\mathrm{SKCE}}_{\mathrm{b}}$ with the mean total variation distance, evaluated on $10^4$ data sets of 250 labeled predictions that are randomly sampled from generative models with $\alpha = (1, \ldots, 1)$ and $\beta = (1/m, \ldots, 1/m)$.

Figure 16: Distribution of $\widehat{\mathrm{SKCE}}_{\mathrm{b}}$ with the mean total variation distance, evaluated on $10^4$ data sets of 250 labeled predictions that are randomly sampled from generative models with $\alpha = (0.1, \ldots, 0.1)$ and $\beta = (1/m, \ldots, 1/m)$.

Figure 17: Distribution of $\widehat{\mathrm{SKCE}}_{\mathrm{b}}$ with the mean total variation distance, evaluated on $10^4$ data sets of 250 labeled predictions that are randomly sampled from generative models with $\alpha = (1, \ldots, 1)$ and $\beta = (1, 0, \ldots, 0)$.

Figure 18: Distribution of $\widehat{\mathrm{SKCE}}_{\mathrm{b}}$ with the mean total variation distance, evaluated on $10^4$ data sets of 250 labeled predictions that are randomly sampled from generative models with $\alpha = (0.1, \ldots, 0.1)$ and $\beta = (1, 0, \ldots, 0)$.

Figure 19: Distribution of $\widehat{\mathrm{SKCE}}_{\mathrm{uq}}$ with the median heuristic, evaluated on $10^4$ data sets of 250 labeled predictions that are randomly sampled from generative models with $\alpha = (1, \ldots, 1)$ and $\beta = (1/m, \ldots, 1/m)$.

Figure 20: Distribution of $\widehat{\mathrm{SKCE}}_{\mathrm{uq}}$ with the median heuristic, evaluated on $10^4$ data sets of 250 labeled predictions that are randomly sampled from generative models with $\alpha = (0.1, \ldots, 0.1)$ and $\beta = (1/m, \ldots, 1/m)$.

Figure 21: Distribution of $\widehat{\mathrm{SKCE}}_{\mathrm{uq}}$ with the median heuristic, evaluated on $10^4$ data sets of 250 labeled predictions that are randomly sampled from generative models with $\alpha = (1, \ldots, 1)$ and $\beta = (1, 0, \ldots, 0)$.

Figure 22: Distribution of $\widehat{\mathrm{SKCE}}_{\mathrm{uq}}$ with the median heuristic, evaluated on $10^4$ data sets of 250 labeled predictions that are randomly sampled from generative models with $\alpha = (0.1, \ldots, 0.1)$ and $\beta = (1, 0, \ldots, 0)$.

Figure 23: Distribution of $\widehat{\mathrm{SKCE}}_{\mathrm{uq}}$ with the mean total variation distance, evaluated on $10^4$ data sets of 250 labeled predictions that are randomly sampled from generative models with $\alpha = (1, \ldots, 1)$ and $\beta = (1/m, \ldots, 1/m)$.

Figure 24: Distribution of $\widehat{\mathrm{SKCE}}_{\mathrm{uq}}$ with the mean total variation distance, evaluated on $10^4$ data sets of 250 labeled predictions that are randomly sampled from generative models with $\alpha = (0.1, \ldots, 0.1)$ and $\beta = (1/m, \ldots, 1/m)$.

Figure 25: Distribution of $\widehat{\mathrm{SKCE}}_{\mathrm{uq}}$ with the mean total variation distance, evaluated on $10^4$ data sets of 250 labeled predictions that are randomly sampled from generative models with $\alpha = (1, \ldots, 1)$ and $\beta = (1, 0, \ldots, 0)$.

Figure 26: Distribution of $\widehat{\mathrm{SKCE}}_{\mathrm{uq}}$ with the mean total variation distance, evaluated on $10^4$ data sets of 250 labeled predictions that are randomly sampled from generative models with $\alpha = (0.1, \ldots, 0.1)$ and $\beta = (1, 0, \ldots, 0)$.

Figure 27: Distribution of $\widehat{\mathrm{SKCE}}_{\mathrm{ul}}$ with the median heuristic, evaluated on $10^4$ data sets of 250 labeled predictions that are randomly sampled from generative models with $\alpha = (1, \ldots, 1)$ and $\beta = (1/m, \ldots, 1/m)$.

Figure 28: Distribution of $\widehat{\mathrm{SKCE}}_{\mathrm{ul}}$ with the median heuristic, evaluated on $10^4$ data sets of 250 labeled predictions that are randomly sampled from generative models with $\alpha = (0.1, \ldots, 0.1)$ and $\beta = (1/m, \ldots, 1/m)$.

Figure 29: Distribution of $\widehat{\mathrm{SKCE}}_{\mathrm{ul}}$ with the median heuristic, evaluated on $10^4$ data sets of 250 labeled predictions that are randomly sampled from generative models with $\alpha = (1, \ldots, 1)$ and $\beta = (1, 0, \ldots, 0)$.

Figure 30: Distribution of $\widehat{\mathrm{SKCE}}_{\mathrm{ul}}$ with the median heuristic, evaluated on $10^4$ data sets of 250 labeled predictions that are randomly sampled from generative models with $\alpha = (0.1, \ldots, 0.1)$ and $\beta = (1, 0, \ldots, 0)$.

Figure 31: Distribution of $\widehat{\mathrm{SKCE}}_{\mathrm{ul}}$ with the mean total variation distance, evaluated on $10^4$ data sets of 250 labeled predictions that are randomly sampled from generative models with $\alpha = (1, \ldots, 1)$ and $\beta = (1/m, \ldots, 1/m)$.

Figure 32: Distribution of $\widehat{\mathrm{SKCE}}_{\mathrm{ul}}$ with the mean total variation distance, evaluated on $10^4$ data sets of 250 labeled predictions that are randomly sampled from generative models with $\alpha = (0.1, \ldots, 0.1)$ and $\beta = (1/m, \ldots, 1/m)$.

Figure 33: Distribution of $\widehat{\mathrm{SKCE}}_{\mathrm{ul}}$ with the mean total variation distance, evaluated on $10^4$ data sets of 250 labeled predictions that are randomly sampled from generative models with $\alpha = (1, \ldots, 1)$ and $\beta = (1, 0, \ldots, 0)$.

Figure 34: Distribution of $\widehat{\mathrm{SKCE}}_{\mathrm{ul}}$ with the mean total variation distance, evaluated on $10^4$ data sets of 250 labeled predictions that are randomly sampled from generative models with $\alpha = (0.1, \ldots, 0.1)$ and $\beta = (1, 0, \ldots, 0)$.

## J.3 Modern neural networks

In the main experiments of our paper discussed in Appendix J.2 we focus on an experimental confirmation of the derived theoretical properties of the kernel-based estimators and their comparison with the commonly used ECE. In contrast to Guo et al. (2017), neither the study of the calibration of different neural network architectures nor the re-calibration of uncalibrated models are the main goal of our paper. The calibration measures that we consider only depend on the predictions and the true labels, not on how these predictions are computed. We therefore believe that directly specifying the predictions in a "controlled way" results in a cleaner and more informative numerical evaluation.

That being said, we recognize that this approach might result in an unnecessary disconnect between the results of the paper and a practical use case. We therefore conduct additional evaluations with different modern neural networks as well. We consider pretrained ResNet, DenseNet, VGGNet, GoogLeNet, MobileNet, and Inception neural networks Phan (2019) for the classification of the CIFAR-10 image data set Krizhevsky (2009). The CIFAR-10 data set is a labeled data set of $32 \times 32$ colour images and consists of 50000 training and 10000 test images in 10 classes. The calibration of the neural network models is estimated from their predictions on the CIFAR-10 test data set. We use the same calibration error estimators and p-value approximations as for the generative models above; however, the minimum number of samples per bin in the data-dependent binning scheme of the ECE estimator is increased to 100 to account for the increased number of data samples.

Figure 35: Calibration error estimates of modern neural networks for classification of the CIFAR-10 image data set.

The computed calibration error estimates are shown in Fig. 35. As we argue in our paper, the raw calibration estimates are not interpretable and can be misleading. The results in Fig. 35 endorse this opinion. The estimators rank the models in different order (also the two estimators of the ECE), and it is completely unclear if the observed calibration error estimates (in the order of $10^{-2}$ and $10^{-4}$!) actually indicate that the neural network models are not calibrated.

Hence to obtain an interpretable measure, we consider different bounds and approximations of the p-value for the calibration error estimators, assuming the models are calibrated. More concretely, we estimate the p-value by consistency resampling of the standard ($\mathbf{C}_{\mathrm{uniform}}$) and the data-dependent

Figure 36: Bounds and approximations of the p-value of modern neural networks for classification of the CIFAR-10 image data set for different calibration error estimators, assuming the models are calibrated.

($C_{\text{data}-\text{dependent}}$) estimator of the ECE, evaluate the distribution-free bounds of the p-value for the estimators $\widehat{\text{SKCE}}_b$ ($\mathbf{D}_b$), $\widehat{\text{SKCE}}_{uq}$ ($\mathbf{D}_{uq}$), and $\widehat{\text{SKCE}}_{ul}$ ($\mathbf{D}_l$) of the SKCE, and approximate the p-value using the asymptotic distribution of the estimators $\widehat{\text{SKCE}}_{uq}$ ($\mathbf{A}_{uq}$) and $\widehat{\text{SKCE}}_{ul}$ ($\mathbf{A}_l$). The results are shown in Fig. 36.

The approximations obtained by consistency resampling are almost always zero, apart from GoogLeNet for the ECE estimator with partitions of uniform size. However, since our controlled experiments with the generative models showed that consistency resampling might underestimate the p-value of calibrated models on average, these approximations could be misleading. On the contrary, the bounds and approximations of the p-value for the estimators of the SKCE are theoretically well-founded. In our experiments with the generative models, the asymptotic distribution of the estimator $\widehat{\text{SKCE}}_{uq}$ seemed to allow to approximate the p-value quite accurately on average and yielded very powerful tests. For all studied neural network models these p-value approximations are zero, and hence for all models we would always reject the null hypothesis of calibration. The p-value approximations based on the asymptotic distribution of the estimator $\widehat{\text{SKCE}}_{ul}$ vary between around $0.18$ for the ResNet18 and $0.91$ for the GoogLeNet model. The higher p-value approximations correspond to the increased empirical test errors with the uncalibrated generative models compared to the tests based on the asymptotic distribution of the estimator $\widehat{\text{SKCE}}_{uq}$. Most distribution-free bounds of the p-value are between $0.99$ and $1$, indicating again that these bounds are quite loose.

All in all, the evaluations of the modern neural networks seem to match the theoretical expectations and are consistent with the results we obtained in the experiments with the generative models. Moreover, the p-value approximations of zero are consistent with Guo et al. (2017)'s finding that modern neural networks are often not calibrated.

## J.4 Computational time

The computational time, although dependent on our Julia implementation and the hardware used, might provide some insights to the interested reader in addition to the algorithmic complexity.

However, in our opinion, a fair comparison of the proposed calibration error estimators should take into account the error of the calibration error estimation, similar to work precision diagrams for numerical differential equation solvers.

A simple comparison of the computational time for the calibration error estimators used in the experiments with the generative models in Appendix J.2 on our computer (3.6 GHz) shows the expected scaling of the computational time with increasing number of samples. As Fig. 37 shows, even for 1000 samples and 1000 classes the estimators $\widehat{\mathrm{SKCE}}_{\mathrm{b}}$ and $\widehat{\mathrm{SKCE}}_{\mathrm{uq}}$ with the median heuristic can be evaluated in around 0.1 seconds. Moreover, the simple benchmark indicates that the evaluation of the estimator of the ECE with data-dependent bins is much slower than of the one with partitions of uniform size.

Figure 37: Computational time for the evaluation of calibration error estimators on data sets with different number of classes versus number of data samples.

## Footnotes

[5]Let $U, V$ be two Hilbert spaces. Then the adjoint of a linear operator $T \in \mathcal{L}(U, V)$ is the linear operator $T^* \in \mathcal{L}(V, U)$ such that for all $u \in U, v \in V$ $\langle Tu, v \rangle_V = \langle u, T^*v \rangle_U$.

[6]For illustrative purposes we present a variation of the original definition of the MMCE by Kumar et al. (2018).

[7]In the words of Vaicenavicius et al. (2019), $\tilde{g}$ is induced by the maximum calibration lens.