[Reviews · NeurIPS 2019]

Reviewer 1



Originality ------------ The present work is a novel unifying view of multiple calibration errors (ECE, MCE, MMCE). The authors relates well their work to the literature as they show how the proposed framework generalize the aforementioned methods. Through this framework, new estimators with interesting theoretical and practical properties are derived. The proposed estimator are all consistent and one of them can be computed in linear time.The estimators are also interpretable in the sense that they can be used for hypothesis testing. Quality --------- To the best of my knowledge the paper is technically sound, and all claims come with quality proofs. The authors backed error bounds by numerical studies. Yet the authors could have included experiments on real neural network (On Calibration of Modern Neural Networks, Guo et Al. 2017). Clarity -------- The paper is well written and organized. The supplementary material embed all the knowledge on Operator-Valued kernel required to understand the proofs. Significance --------------- Calibration of classification model is a very important topic for industries dealing with critical applications. Moreover the present work brings a new theoretical unifying view on calibration errors. I have read the author response and changed my score from 7 to 8.

Reviewer 2



***** Post-rebuttal update: after reading the authors' feedback, I confirm my positive evaluation of this paper. ***** I would like to thank the authors for their submission. Summary The paper presents a novel unified theoretical framework and new measures for the calibration properties of multi-class classifiers, which generalize commonly used ones. Estimators for the proposed measures, based on vector-valued RKHS, are then proposed. The statistical properties of such estimators are theoretically characterized (including proofs), and statistical tests associated to the estimators are presented. Finally, the properties of the proposed estimators are exhaustively validated in supporting simulated experiments. Originality The proposed ideas are novel in the context of calibrated multi-class classification. The proposed methods (e.g. the definition of KCE and the estimators) draw tools from matrix-valued kernel methods and kernel two-sample tests, which are appropriately referenced. To my knowledge, this area of research has a rather scattered coverage in the literature. The paper presents potentially high-impact novel contributions and provides a much-needed rigorous unifying view on the topic. To my knowledge, other relevant works in the area are correctly referenced and differences clearly stated. Quality I have found the paper to be of rigorous technical soundness. Definitions and statements are all clear and quantities properly introduced. Statements are remarkably supported by both theoretical statements, including full proofs, and exhaustive, well-designed experiments on synthetic data. Clarity The motivation, context, literature review, problem statement, theoretical claims and experiments are all delivered with excellent clarity and well-organized. The paper is smooth, polished and pleasant to read. Significance As stated in the "Contributions" section, I deem the overall significance of this work as high, from both the theoretical and practical perspectives. Developing rigorous tools for characterizing the quality of predictive models' confidence predictions is an important priority for the field, and I have little doubt about this paper representing an important step forward in this direction. Typos and minor comments: - Capitalize first letters in title and section headings - L15: patients.Since --> patients. Since - L17: increasing the training data set is... --> increasing the training data set's size is... - L22: Thus, - L24: uncertainty, this - L45: complementary - L46: miscalibrated, - L56: Recently, - L58: ..., 0.3) since --> ..., 0.3), since - L70: detail, - Eq. (1): is the conditioning on $max_y g_y (X)$ right, or could it just be on $g(X)$? - L104, L114, L121, ... : Thus, - L104, L117, L118, ... : eq. --> Eq. - L159: a general calibration measure of strong calibration --> a general measure of strong calibration - L180: setting, - L257, ...: fig. --> Fig. - L310: Consider citing as: Carmeli, C., De Vito, E., Toigo, A., & Umanitá, V. (2010). Vector valued reproducing kernel Hilbert spaces and universality. Analysis and Applications, 8(01), 19-61.

Reviewer 3



Summary: In this paper the authors propose a unifying framework for testing multi-class classification calibration, i.e. if a classifier gives a probabilistic output, how close (or if, as a statistical test) are these outputs to the actual probabilities given the classifier (the actual definition is actually a bit more subtle, but this is the general idea). To do this they introduce a framework involving an integral probability metric. They propose using a "matrix kernel" RKHS as a nice way of controlling the function which they maximize in the probability metric. With this kernel setup they have a nice closed form expression for finding the calibration error and some finite and asymptotic bounds describing its behavior. Finally the authors perform experiments where they empirically evaluate the distribution of the test statistic, as well as its type I and II errors, and compare to a classic method. Overview: Overall the paper reads well and is fairly clear. The theory looks good and I see the kind of theorems and bounds I would expect to find in this kind of paper. The experiments seem good, although I think they could consider more competitor techniques (assuming they exist, I am not terribly familiar with this specific topic). Perhaps it would also be interesting to see how the test performs on a real world dataset; however this test statistic does not depend on the value of X, just on Y values and the prediction output distributions, so perhaps a real world dataset doesn't really add much to what the authors have done already. The supplementary material is quite extensive and contains quite a bit of good, mathematically correct, theory along with more experiments. Potential Issues: It is unclear to me why "matrix valued kernels" are necessary for this test. I haven't personally derived anything but it seems likely that one could do this sort of test using standard kernels. Perhaps the "matrix valued kernel" arises naturally from the obvious test statistic, but it would be nice to know why it is necessary for this concept to be introduced. I am unsure how significant this paper is. The fact that the paper only provides a statistical test is a bit concerning. Again I am not familiar with this particular problem, perhaps it is difficult and/or important enough that this paper is important. I found l.213-216 a bit too mysterious for my liking. Is it not possible for other estimators to estimate (3). Does this method allow (3) to become tractable, or are you simply observing that one should keep in mind that test statistics come from (3). Verdict: While I'm not entirely sure of the significance of the topic the paper is well written and the research seems high quality, so I recommend that it should be accepted. Small errors or potential improvements: l.6-8 This sentence is grammatically incorrect, or at least strange. Maybe something like "We present a new method based on matrix-valued kernels which offers consistent and unbiased..." l.15 Missing a space after the period l.18 "would be" should be "is" l.35 maybe this is nomenclature for the topic, but I'm not totally sure what "subjective" means here l.38-47 I found this paragraph confusing while reading. I think it could be improve by using a concrete mathematical expression or two. I didn't really understand the problem until I got to (1) and (2) l.52 "were" should be "have been" l.133 "matrix-valued" kernel should be \emph l.139-148 Its difficult for me to figure out what here is established theory and what is stuff that you have developed and have just left underived. For example is "universal kernel" totally analgous to the standard definition or is it a bit different? Also how does the Micchelli and Pontil 2005 relate to this? Did they actually introduce the matrix valued kernel or just a vector valued one which you use as the basis for the matrix one. l.161 its not clear to me why the CE has to be infinity if its not 0. If F only contains bounded functions that it seems to me that maximizing over F should remain bounded. I wonder if you are already adopting some of the intuition from the RKHS function class (e.g. assuming F is a vector space) l.219 "greater or equal than" should be "greater than or equal to" I find the P[...] notation a bit strange, I think it is definitely less standard than P(...) l.225 "weak" should be "loose"

[Author Response · NeurIPS 2019]

We are very happy to hear that all reviewers found the paper interesting and well written. We thank the reviewers for all the constructive feedback on our paper and for all the suggestions for future work. We will naturally take all comments into account when revising the paper. Below we provide point-by-point responses to selected comments.

**All reviewers commented on that it would have been interesting to see results on real-world datasets.**

We intentionally did not include any evaluations on benchmark datasets such as CIFAR-10 or ImageNet, since we wanted to focus on an experimental confirmation of the derived theoretical properties of the kernel-based estimators and the illustration of its advantages over the commonly used expected calibration error (ECE). In contrast to Guo et al., we did not want to make any claims about the calibration of different neural network architectures, nor the re-calibration of uncalibrated models.

As pointed out by Reviewer #3, the calibration measures that we consider only depend on the predictions (and true labels), not on how these predictions are computed. We therefore believe that directly specifying the predictions in a "controlled way" results in a cleaner and more informative numerical evaluation.

That being said, we recognize that his approach might have resulted in an unnecessary disconnect between the results of the paper and a practical use case. We have therefore conducted additional evaluations with different neural networks such as DenseNet, ResNet, GoogleNet, Inception, and VGGNet trained on the CIFAR-10 dataset, using the same binning and kernel choices as in our submission. As we have argued in our submission, the raw calibration estimates are not interpretable and can be misleading, and hence we have only considered the proposed approximations of the probability of falsely rejecting a calibrated model. The distribution-free bounds are typically between 0.99 and 1, indicating again their weakness, whereas the bounds based on the asymptotic distribution of linear unbiased SKCE yield values between 0.09 (for ResNet-34) and 0.91 (for GoogleNet). On the other hand, the approximations obtained by consistency resampling, both with uniform and data-dependent binning, are almost always 0 (apart for GoogleNet with uniform binning). It seems the quadratic bound with (only) 100 bootstrap samples for the asymptotic distribution yields also 0 for all models. We will add these additional experimental results to the revised supplementary material.

**Reviewer #1**

"it could be useful to study the impact of the chosen kernel and its hyperparameters"

Indeed, the impact of the kernel and its hyperparameters on the estimators and, in particular, on the bounds of the type I error is an important research question. Apart from the choice of the kernel bandwidth, we had refrained from discussing it in our initial submission. In our opinion, this question deserves a more exhaustive study than, what we felt, would have been possible in this work. We will state the need and importance for future research in this area more clearly.

**Reviewer #2**

"Consider reporting computational time for the proposed estimators and experimental setup details"

We agree that the computational time, although dependent on our Julia implementation and the hardware used, might provide some insights to the interested reader in addition to the algorithmic complexity. However, in our opinion, a fair comparison of the suggested estimators and p-value approximations should take into account the error of the methods as well, similar to work precision diagrams for numerical differential equation solvers. In our case, one could quantify the bias and variance of the estimators and the p-value approximations. A simple comparison of the methods (with the same kernel and binning choices as in our submission) for fixed numbers of classes and varying number of samples has shown the expected scaling of the computation time with increasing number of samples and revealed that even for 1000 samples and 1000 classes the biased SKCE and the quadratic unbiased SKCE can be evaluated in around 0.1 sec. As a comparison, for this setting, the evaluation of the linear unbiased SKCE takes around $10^{-4}$ sec, of the ECE with bins of uniform width around $10^{-3}$ sec, and of the ECE with data-dependent bins around 0.1 sec.

**Reviewer #3**

"I found l.213-216 a bit too mysterious ... Is it not possible for other estimators to estimate (3). Does this method allow (3) to become tractable, or are you simply observing that one should keep in mind that test statistics come from (3). "

We simply mean that the statistical tests are designed to test if (3) does not hold. Other estimators are indeed possible, but establishing deviation inequalities such as our Lemmas 2 and 3 has to be done on a case-by-case basis, and this could very well prove to be difficult in many cases.

**References**

Chuan Guo, Geoff Pleiss, Yu Sun, and Kilian Q. Weinberger. On calibration of modern neural networks. In *Proceedings of the 34th International Conference on Machine Learning*, volume 70 of *Proceedings of Machine Learning Research*, pages 1321–1330. PMLR, 08 2017.


[Meta-Review · NeurIPS 2019]

The paper brings forward a new rigorous framework for the calibration of multi-class classification models. The reviewers found the contributions to be significant and original and the paper well written. The authors clarified the unclear points in their response.